J Physiol 603.23 (2025) pp 7603–7625

7603

# Muscle regeneration is improved by hot water immersion but unchanged by cold following a simulated musculoskeletal injury in humans

Valentin Dablainville[1,2] (ID), Adèle Mornas[3,4,5] (ID), Tom Normand-Gravier[2,6] (ID), Maha Al-Mulla[1], Emmanouil Papakostas[7] (ID), Bruno Olory[7] (ID), Theodorakys Marin Fermin[7,8] (ID), Frantzeska Zampeli[7,9] (ID), Nelda Nader[1], Marine Alhammoud[7,10] (ID), Freya Bayne[11] (ID), Anthony M. J. Sanchez[6] (ID), Marco Cardinale[1,12,13] (ID), Robin Candau[2], Henri Bernardi[2] (ID) and Sébastien Racinais[1,2,3,14] (ID)

[1]Aspetar Orthopaedic and Sports Medicine Hospital, Research and Scientific Support Department, Doha, Qatar
[2]DMEM, University of Montpellier, INRAE, Montpellier, France
[3]French Institute of Sport (INSEP), Laboratory Sport, Expertise and Performance (EA 7370), Paris, France
[4]University Paris Cité, Paris, France
[5]Montreal Heart Institute, Montréal, Quebec, Canada
[6]Faculty of Sports Sciences, Laboratoire interdisciplinaire Performance Santé Environnement de Montagne (LIPSEM), UR4640, University of Perpignan Via Domitia (UPVD), Font-Romeu, France
[7]Aspetar Orthopaedic and Sports Medicine Hospital, Surgery Department, Doha, Qatar
[8]Centro Médico Profesional Las Mercedes, Caracas, Venezuela
[9]Hand-Upper Limb-Microsurgery Department, General Hospital KAT, Kifisia, Greece
[10]Inter-University Laboratory of Human Movement Biology (EA 7424), Savoie Mont Blanc University, Chambéry, France
[11]Sport and Exercise Science Research Centre, School of Applied Sciences, London South Bank University, London, UK
[12]Institute of Sport, Exercise and Health, University College London, London, UK
[13]Department of Sport, Exercise and Rehabilitation, Faculty of Health and Life Sciences, Northumbria University, Newcastle upon Tyne, UK
[14]CREPS Montpellier Font-Romeu, Environmental Stress Unit, Montpellier, France

The peer review history is available in the Supporting Information section of this article (https://doi.org/10.1113/JP287777#support-information-section).

The Journal of Physiology

H. Bernardi and S. Racinais contributed equally to this work.

The Journal of Physiology

**Abstract figure legend** Thirty-four participants underwent muscle damage (electrically stimulated eccentric contractions) triggering regenerative processes following myofiber necrosis. Participants were treated for 10 days cold (CWI) (15 min, 12{degree sign}C), thermoneutral (TWI) (30 min, 32{degree sign}C) or hot (HWI) (60 min, 42{degree sign}C) water immersion. HWI induced lower perceived muscle soreness than TWI and lower levels of circulating creatine-kinase and myoglobin than TWI and CWI, without differences in force. HSP 27 and 70 were significantly increased by HWI at D11 and blunted by CWI. HWI upregulated IL-10 at D11 and blunted the increase in NF-$\kappa$B, suggesting an earlier shift from pro to anti-inflammatory phase. In conclusion, our results support the use of hot water immersion but not cold, to improve muscle recovery following an injury.

**Abstract** Cryotherapy is a popular strategy for the treatment of skeletal muscle injuries. However, its effect on post-injury human muscle regeneration remains unclear. In contrast, promising results recently emerged using heat therapy to facilitate recovery from muscle injury. This study aimed to examine the effect of three different thermal treatments on muscle recovery and regeneration following a simulated injury in humans. Thirty-four participants underwent a muscle damage protocol induced by electrically stimulated eccentric contractions triggering regenerative processes following myofibre necrosis. Thereafter, participants were exposed to daily lower body water immersion for 10 days in cold (CWI, 15 min at 12°C), thermoneutral (TWI, 30 min at 32°C) or hot water immersion (HWI, 60 min at 42°C). Muscle biopsies were sampled before and at +5 (D5) and +11 (D11) days post-damage. None of the water immersions differed in recovery of force-generating capacity ($P = 0.108$). HWI induced a lower perceived muscle pain than TWI ($P = 0.035$) and lower levels of circulating creatine kinase ($P \leq 0.012$) and myoglobin ($P < 0.001$) than TWI and CWI. Contrary to our hypothesis, CWI did not improve perceived muscle pain or reduce circulating markers of muscle damage ($P \geq 0.207$). Expression of heat shock proteins 27 and 70 was significantly increased in HWI ($P < 0.038$) at D11 and appeared blunted using CWI. Furthermore, nuclear factor-$\kappa$B expression significantly increased in all conditions except HWI, while interleukin-10 was upregulated only in HWI at D11 ($P = 0.014$). In conclusion, our results support the use of HWI but not cold, to improve muscle regeneration following an injury.

(Received 30 September 2024; accepted after revision 23 April 2025; first published online 29 May 2025)

**Corresponding author** S. Racinais: CREPS Montpellier, 2 Av. Charles Flahault, 34090 Montpellier, France. Email: sebastien.racinais@creps-montpellier.sports.gouv.fr

## Key points

- Cryotherapy and heat therapy are popular strategies in the treatment of skeletal muscle injury; however, existing literature is equivocal, and their effects on human muscle regeneration remain unknown.
- We investigated the effect of three thermal treatments (cold water immersion (CWI): 15 min at 12°C; thermoneutral water immersion (TWI): 30 min at 32°C; or hot water immersion (HWI): 60 min at 42°C) performed daily for 10 days following electrically stimulated eccentric muscle damage inducing regenerative mechanisms.
- CWI did not improve chronic perceived muscle pain nor reduce circulating markers of muscle damage.
- HWI limited chronic perceived pain and circulating markers of muscle damage, potentially influenced inflammatory mechanisms, and increased the expression of heat shock proteins.
- HWI appears more beneficial than CWI in improving muscle regeneration after a muscle injury.

## Introduction

Muscle regeneration, defined by the myogenesis events following myofibre necrosis, is a complex biological process restoring muscle integrity after muscle injury and damage (Grounds, 2014). While different forms and severities of muscle injury are associated with different healing kinetics, the processes involved in muscle

regeneration share many similarities (Mackey & Kjaer, 2017). Muscle regeneration generally involves a succession of interrelated phases: degeneration, inflammation, regeneration and remodelling necessary to restore functional tissue (Forcina et al., 2020). Furthermore, other mechanisms, such as angiogenesis, are highly involved and essential for successfully completing healing of injured tissues (Jacobsen et al., 2023). Among the solutions proposed to facilitate muscle healing, local thermal therapies, which involve tissue cooling or heating, represent some of the most popular strategies (Frery et al., 2023; McGorm et al., 2018).

Cold water immersion (CWI) is a popular post-injury cryotherapeutic intervention commonly used by athletes to accelerate recovery from muscle damage and delayed onset of muscle soreness (Bleakley & Davison, 2010a; Hohenauer et al., 2015; Kwiecien & McHugh, 2021). Also, local cryotherapy, often defined as the removal of heat from a targeted tissue, has been largely advised as an acute, post-injury intervention to decrease pain and limit secondary injury, inflammation, swelling and haematoma (Jarvinen et al., 2019; Kwiecien & McHugh, 2021). However, as recently highlighted, there is currently a lack of evidence supporting the alleged benefits of cryotherapy on skeletal muscle healing, and to our knowledge, no human study has explored the effect of cryotherapy on muscle regeneration (Bleakley et al., 2004; Collins, 2008; Hubbard et al., 2004; Normand-Gravier et al., 2024; Racinais et al., 2024). In animal models, a recent study performed in rats showed that applying ice on limited muscle necrosis (representing <10% of the myofibre fraction) could facilitate muscle regeneration by attenuating degenerative and inflammatory processes and accelerating the accumulation of myogenic cells (Nagata et al., 2023). However, on larger muscle injuries (necrosis representing >10% of the myofibre fraction), most of the literature on animal models suggests that post-injury cryotherapy may impair and delay muscle regeneration mainly through blunted inflammatory mechanisms (Kawashima et al., 2021; Miyakawa et al., 2020; Miyazaki et al., 2023; Singh et al., 2017; Takagi et al., 2011).

In contrast, localised passive heating has emerged as a potential strategy to enhance muscle healing in humans (McGorm et al., 2018; Normand-Gravier et al., 2024). During recovery from eccentrically induced muscle damage, heat stress was shown to promote the expression of angiogenic factors (Kim et al., 2019) and to mitigate the post-exercise decrease in force-generating capacity (Sautillet et al., 2023, 2024). A sufficient level of heat stress also upregulates protein synthesis signalling in healthy individuals (Ihsan et al., 2020; Kakigi et al., 2011), and limits muscle atrophy in immobilised healthy individuals (Hafen et al., 2019; Labidi et al., 2024). Although the effect of heat therapy on human muscle healing remains unclear, results from animal model studies suggest improved muscle regeneration with heat exposure. In fact, a single post-injury local heat application (hot water immersion (HWI) or hot packs applied to the limbs) was shown to accelerate muscle fibre cross-sectional area recovery (Shibaguchi et al., 2016; Takeuchi et al., 2014), accelerate macrophage infiltration (Takeuchi et al., 2014) and induce a greater number of satellite cells within the damaged muscle (Hatade et al., 2014; Oishi et al., 2009, 2015; Takeuchi et al., 2014).

Despite the large use of thermal therapies in the treatment of acute muscle injuries, especially cryotherapy, to our knowledge, no evidence currently supports their efficacy in human muscle healing. Indeed, technical and ethical challenges limit the investigation of post-injury regenerative mechanisms, and thermal therapy effects have been mostly studied using exercise-based models of voluntary eccentric contractions that do not, or only rarely, induce myofibre necrosis and subsequent muscle regeneration. In contrast, electrically stimulated eccentric contraction models induce large myofibre necrosis and appear to be an effective solution to simulate muscle injury and study muscle regeneration in humans (Crameri et al., 2007; Mackey & Kjaer, 2017).

Therefore, the aim of the current study was to determine the effects of CWI and HWI on muscle regeneration in humans. For this purpose, muscle force, muscle pain, blood samples and muscle biopsies were obtained following an electrically stimulated eccentric muscle damage to determine the recovery of muscle function along markers of inflammation and remodelling under different treatments. Based on the existing literature

**Valentin Dablainville** is a PhD student in muscle physiology in the laboratory Muscle Dynamic and Metabolism (DMEM) from the University of Montpellier, France. He is working as researcher in Aspetar sport medicine hospital, Qatar. His PhD focuses on the use of thermal therapies in the recovery from muscle injuries. **Sébastien Racinais** obtained his PhD in 2004 and is currently working at the institute sports of Montpellier, in France (CREPS). Prof Racinais has successfully guided numerous professional and national teams in developing environmental training, and is collaborating with international sports federations. He also leads the European Network in Sports Sciences (ENSS) in environmental physiology; is the Chair of the IOC Adverse Weather Impact expert working group for the Olympic Games Tokyo 2020 and Paris 2024; and a member of the Medical and Scientific Commission Games Group for the protection of athlete's health. His research is currently developing heat therapy for muscle rehabilitation.

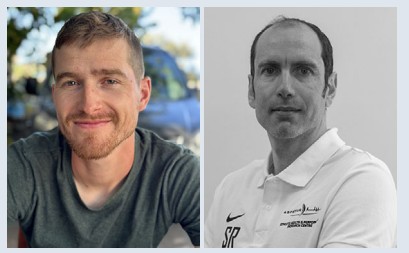

**Table 1. Participants and muscle damage data**

| Condition | CWI ($n$ = 12) | TWI ($n$ = 11) | HWI ($n$ = 11) |
|---|---|---|---|
| Age (years) | 33.2 ± 2.6 | 35.2 ± 4.3 | 32.6 ± 3.5 |
| Weight (kg) | 88.7 ± 7.3 | 82.0 ± 11.3 | 84.9 ± 14.3 |
| Height (cm) | 180.5 ± 3.8 | 180.7 ± 5.8 | 181.3 ± 7.1 |
| MVC (kg) at PRE | 92.5 ± 15.3 | 88.3 ± 11.9 | 87.0 ± 20.6 |
| Quadriceps stimulation intensity during muscle damage protocol (mA) | 113.5 ± 14.0 | 111.2 ± 14.4 | 101.2 ± 22.6 |
| Average work per set per weight in kg during muscle damage protocol (J/kg) | 19.1 ± 1.5 | 19.2 ± 1.5 | 18.0 ± 1.5 |

we hypothesised that: (1) CWI would decrease the overall pain perception but would delay the inflammatory processes and muscle regeneration, and (2) HWI would facilitate the inflammatory processes and facilitate muscle regeneration.

## Methods

### Ethical approval

This study was approved by the Aspetar Orthopaedic and Sport Medicine Hospital scientific committee (no. ASC/0000245/ak) and by Aspire Zone Foundation institutional review board (no. F202202032) and registered on ClinicalTrials.gov (Ref: NCT05506514). The study conformed to the standard of the *Declaration of Helsinki*, and written informed consent was obtained from all participants before any procedure.

### Participants

Thirty-six healthy men (age: 33 ± 3.5 years; weight: 85.3 ± 10.9 kg, height: 180.8 ± 5.6 cm) were initially recruited to participate in this study. However, two participants were excluded from the analyses as they had unsupervised physical activity during the experiment. Thus, 34 participants were retained, and their data were analysed (Table 1). As computed using G*Power statistical analyses software (version 3.1.9.7, Heinrich-Heine-Universität Düsseldorf, Düsseldorf, Germany) for an output measured four times in three groups, a minimal sample size of 27 participants (9/group) was required with a 0.05 $\alpha$-error, and a 0.9 $\beta$-error (Faul et al., 2009). The sample size was increased to 12/group to compensate for potential participant drop-out.

The participants were healthy, mostly former athletes, working as fitness instructors during the time of the study and maintaining a recreational level of training. The inclusion criteria were being healthy, aged between 18 and 45 years old, and having no neuromuscular disorder, injury or contraindication to exercise. Circulating creatine-kinase and myoglobin levels were assessed during the week preceding the start of the experiment to ensure that all participants had values within the normal ranges (31–936 U/L for CK and 0–72 μg/l for myoglobin).

Participants were asked to refrain from training and to avoid any form of exercise involving the lower limbs the week preceding the experiment and throughout the experimental protocol. Exclusion criteria included prescribed medication for chronic medical conditions. Furthermore, participants were instructed to refrain from taking any pain killers or anti-inflammatories. All participants attended two nutritional consultations conducted by a sports dietician (N.N.). They first received an initial face-to-face education with the use of a video and food models at the familiarisation visit. This was followed by individualised meal plans provided prior to the first biopsy (PRE2). The nutritional interventions promoted a standardised energy and protein intake without setting absolute targets, encouraging consistency in intake and meal frequency. Participants were advised to consume at least three servings of 20 g of animal protein daily during the study. Participants were instructed, both during the nutritional interventions and on several other occasions, by the research team not to consume any supplements (e.g. whey proteins, creatine), as well as alcohol and pain medication during the study period. A blood test was performed within a week before the start of the experiment to verify the absence of kidney or liver dysfunction. Due to the burden associated with simulated muscle injury and repeated muscle biopsies, participants were financially compensated for their participation in this study.

### General procedure

An overview of the study design is presented in Fig. 1. During recruitment, participants underwent a familiarisation session, which included an introduction to the tests and a shortened version of the muscle damage protocol (four contractions 30°/s and four contractions 180°/s). Following the familiarisation, baseline testing

(PRE1 and PRE2), including maximal voluntary isometric force, blood sample, perceived pain, and a first muscle biopsy from the left thigh (control leg), was carried out the week preceding the muscle damage protocol. On Day 0, all participants were exposed to an electrically stimulated eccentric protocol to induce muscle injury. Participants were then counterbalanced to one of the three intervention groups: (1) CWI ($n = 12$), (2) thermoneutral water immersion (TWI, $n = 11$) or (3) HWI ($n = 11$). Maximal voluntary isometric contraction (MVC) force and body weight were similarly distributed in the treatment groups. The first thermal intervention was performed 1 h after the completion of the muscle damage protocol (Fig. 1). For the 11 days following the muscle damage protocol, participants reported to the lab daily at the same time of the day to complete testing followed by their assigned thermal intervention. Blood samples were also collected at PRE1, D4 and D8. Muscle biopsies were sampled at PRE2, D5 and D11, between 06.00 and 08.00 h, with the participants fasting.

## Muscle damage protocol

The electrically stimulated eccentric muscle damage protocol used in this study was adapted from previous studies (Crameri et al., 2007; Karlsen et al., 2020; Mackey et al., 2016). Similarly to previous experiments, the quadriceps muscle was chosen to perform the muscle damage protocol because of the ease of performing a muscle damage protocol and collecting biopsy sample compared to other muscle. Participants were seated on a Biodex System 3 isokinetic dynamometer (Biodex Medical Systems Inc., Shirley, NY, USA) set for leg extension/flexion movement with an adopted range of motion from 90° to 10° (with 0° as fully extended position of the leg) with their trunk kept upright. Three stimulation electrodes (Dura-stick plus, CefarCompex, Malmö, Sweden) measuring 9 × 5 cm were placed perpendicularly to the right quadriceps muscle, one proximally across the vastus lateralis (∼5 cm from the greater trochanter) and two distally on the vastus medialis and vastus lateralis muscles (respectively ∼3 and ∼5 cm from the top of the patella). The stimulations were delivered by a constant current stimulator (Digitimer DS7AH, Welwyn Garden City, UK) with continuous impulse trains (frequency 40 Hz, 200 μs pulse duration, 65–135 mA intensity, 400 V). The electrical stimulation began at the initiation of downward movement of the dynamometer lever arm and was turned off at the end of this downward movement. To standardise the stimulation intensity across all participants, the intensity was set as the intensity eliciting 25% of knee extensor MVC during the familiarisation session. Participants were attached to the seat and carefully instructed not to produce any voluntary muscle contraction during the stimulation phase of the exercise protocol. The exercise protocol consisted of a total of 200 electrically stimulated eccentric contractions of the right thigh muscles, the leg was stretched/flexed while the right quadriceps muscle was stimulated for: (1) five sets of 20 repetitions at an angular velocity of 30°/s followed by (2) five sets of 20 repetitions at 180°/s. A 30-s break between each set and a 5-min break between the slow and fast contractions were given. The total work value calculated by the Biodex System was recorded for each participant.

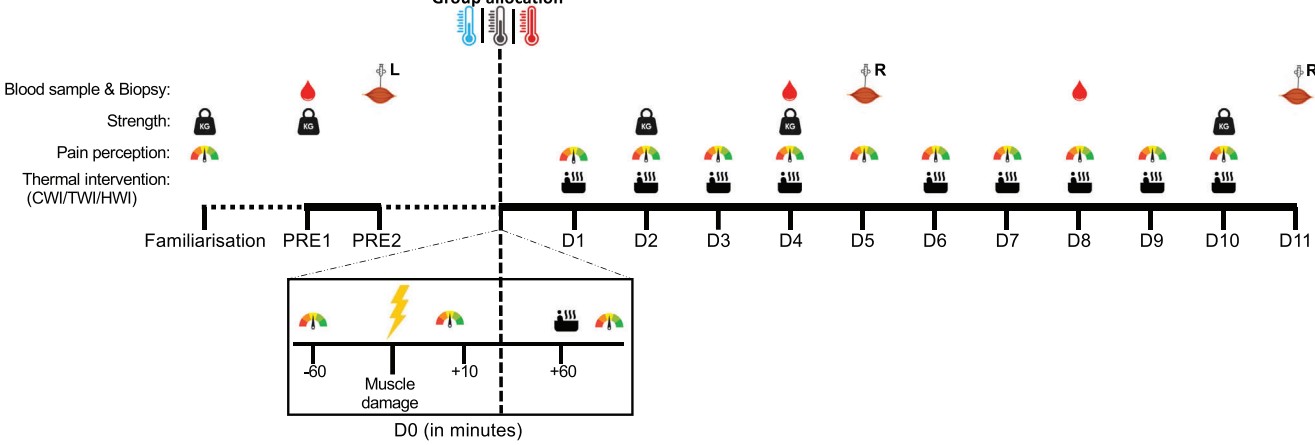

**Figure 1. Schematic overview of the study**
Participants were allocated to their thermal intervention group on day (D) 0 after completing the muscle damage protocol of the right leg. The thermal intervention consisted of 15 min at 15°C cold water immersion (CWI) or 30 min at 32°C thermoneutral water immersion (TWI) or 60 min at 42°C hot water immersion (HWI). PRE biopsy was sampled from left leg and D5/D11 biopsies were sampled from right leg.

## Thermal interventions

Thermal interventions were applied by immersing the legs up to the waist in a water bath (IcePod 550L, icool, Miami, Australia) connected to an external pump heater/cooler (compact XP dual Temp, icool, Miami, Australia) regulating water temperature. Thermal interventions consisted of a 60-min water immersion at 42°C for the HWI group, a 15-min water immersion at 12°C for the CWI group and a 30-min immersion at 32°C for the TWI group (Bleakley & Davison, 2010a; McGorm et al., 2018; Rodrigues et al., 2020; Versey et al., 2013). The temperatures and duration of thermal intervention were chosen based on the published literature as well as typical practices and recommendations made to athletes (Kwiecien & McHugh, 2021; McGorm et al., 2018; Rodrigues et al., 2020). A sham condition using TWI was used (Versey et al., 2013). Each participant performed a total of 10 sessions, except for three participants who completed nine sessions and one participant who completed eight sessions due to external constraints. The first thermal interventions were initiated 1 h after the muscle damage protocol for all the groups. From D1 to D10 post-muscle damage, daily thermal interventions were performed immediately after daily testing measurement (except during biopsy sampling day on D5, where no bathing was performed to avoid wound contamination). Water bath temperatures were constantly monitored by a medical precision thermometer (DM852, Ellab, Hillerød, Denmark) and adjusted as necessary. Room temperature was maintained at 24°C and humidity was 40%. Participants were free to drink *ad libitum* and cooling solutions such as cold water, ice or iced towels to apply to the upper body were proposed to HWI group participants to limit the increase in core temperature and maintain an acceptable thermal comfort. The thermal interventions used in this study were designed to modify muscle temperature and local blood flow, with minimal changes in core body temperature.

## Tests and measures

**Muscle pain.** Muscle pain was assessed at PRE, immediately post-muscle damage (post-exercise at D0), after the first bath (post-immersion at D0), and then daily at rest as the first measurement taken each day. A visual analogue scale (VAS) was used to assess pain perception, similarly to previous studies using electrically stimulated eccentric muscle damage (Crameri et al., 2004, 2007; Mackey et al., 2016). The VAS used in this study was graduated from 0 (no pain) to 20 (worst pain). No graduation was visible by the participant during the measurement. Two different methods were used to assess muscle pain: (1) self-palpation: participants self-palpated their right quadriceps by pressing with their fingertips, starting from the proximal end and moving towards the knee and rated their muscle pain using the VAS; and (2) dynamic squat: participants performed three squats down to a chair (i.e. ∼90 degrees knee angle), only touching the chair and immediately standing, and rated their pain using the same VAS.

**Strength measurements.** The isometric knee extensor force of the right leg was measured using a high-rigidity custom-made dynamometric chair. Participants were seated with both the hip and the knee at 100° (full extension representing 180°), secured with straps around the chest and hip and with the ankle tied to a strain gauge (Captels, St Mathieu de Treviers, France) connected to a stationary bench. Participant positioning was recorded to ensure consistency across tests. Force signal was recorded at a sampling frequency of 2000 Hz using Biopac MP35 hardware (Biopac Systems Inc., Goleta, CA, USA) and its dedicated software (BSL pro 3.7.2, Biopac systems Inc.).

MVC was assessed four times during the protocol: at PRE and 2, 4 and 10 days after the muscle damage protocol. Participants performed 10 warm-up contractions of the quadriceps, each lasting 3–5 s, gradually increasing the force of each contraction up to 80% of their estimated force. Then, after a 2-min rest period, they were instructed to 'produce the highest force possible during 3–5 s' and were encouraged by the investigators. During analysis, the highest value from each session was selected, and the force plateau over 600 ms was averaged to calculate the MVC value for each session.

**Blood samples.** Blood samples were collected from the antecubital vein by a phlebotomist at PRE1 (during the week before the damaging exercise), 4 days and 8 days after the damaging exercise. All samples were centrifuged 10 min at 1,885 g. Creatine kinase (CK) plasma levels were measured by the hospital laboratory department in the 2–3 h following sampling using the Dimension Vista 500 chemistry system with Siemens CKI Flex reagent cartridge (Siemens, Munich, Germany, limit of detection 14,000 U/L). For myoglobin (Mb), plasma was separated from red blood cells, stored at a temperature of 2–8°C and shipped in a cool box to an accredited partner laboratory (Bioscientia, Ingelheim am Rhein, Germany) where specimens were analysed using Siemens BN II and N latex Myoglobin reagent (Siemens Healthineers, Erlangen, Germany). The turnaround time for this test was within 3 days.

**Muscle and body core temperature.** Vastus lateralis intra-muscular temperature was measured once per participant, immediately after the thermal intervention, using a sterile thermo-sensor needle (MKA08050-A, Ellab, Hillerød, Denmark). Temperature measurements

were taken under local anaesthesia (EMLA cream 5% lidocaine/prilocaine, Aspen, St Leonards, Australia) from the unexercised vastus lateralis muscle belly. Muscle temperature was assessed in the unexercised leg to avoid interfering with the regenerative processes of the exercised leg and to limit pain levels experienced by the participants, as the damaged muscle remained painful until the end of the study period. Measurements were recorded at 3, 2 and 1 cm depth under the skin with the thermo-sensor needle left at each depth for approximately 4 s to ensure stable temperature value.

Core temperature was also monitored on one occasion using ingestible pill thermometers (e-Celsius, Bodycap, Hérouville Saint-Clair, France). The core temperature pill was ingested with water at least 4 h before the start of HWI and 1 h before the start of CWI or TWI. While the HWI group was allowed to drink *ad libitum* during the water immersion, the CWI and TWI groups were instructed to refrain from eating or drinking until core temperature measurement was achieved. The e-Celsius pills contain a factory-calibrated sensor, ensuring ±0.1°C precision, and were verified by the supplier in a temperature-controlled bath to ensure their conformity to the targeted temperature range of 36–41°C.

**Muscle biopsies.** Three muscle biopsies were collected by orthopaedic surgeons from the vastus lateralis muscles for each participant during the protocol. The first biopsy was collected from the left unexercised leg during the week preceding the muscle damage. The second and third biopsies were collected from the right leg, respectively, 5 and 11 days after the muscle damage protocol. Local anaesthesia (lidocaine 1%) was applied around the thickest place of the mid-portion of the vastus lateralis. After disinfection of the skin, a small incision was made in the skin and the fascia, and a 5 mm Bergstrom biopsy needle with manual suction (Stille, Torshälla, Sweden) was inserted into the muscle. Muscle samples were quickly cleared of fat, connective tissue and blood excess and divided into two sections. Samples for histological analysis were embedded into Tissue-Tek optimal cutting temperature compound and frozen in isopentane pre-cooled with dry ice at −80°C. The remaining samples were immediately frozen in liquid nitrogen for western blot analysis. All samples were stored at −80°C for 6 months to 1 year before being analysed. Biopsies were sampled between 06.00 and 08.00 h, and participants were instructed to fast on biopsy days. Additionally, meals (breakfast, lunch, snacks and dinner) were individually planned by the hospital's sports dietician and kept constant each day preceding a biopsy. Protein intake was maintained at not less than 20–30 g per meal.

**Western blot.** Frozen muscles from biopsies were powdered under liquid nitrogen, and a known quantity of powdered muscles was homogenised in lysis buffer containing 1% Triton X-100, 20 mM Mops, pH 7.0, 2 mM EGTA, 5 mM EDTA, 30 mM NaF, 60 mM $\beta$-glycerophosphate, 20 mM sodium pyrophosphate, 1 mM sodium orthovanadate, 1 mM dithiothreitol, protease inhibitor (Complete mini, ref 11836153001, Roche, Basel, Switzerland). Homogenates were sonicated four times for 10 s on ice to shear nuclear DNA. After 30 min under agitation, samples were centrifuged at 10,000 $g$ for 15 min at 4°C and protein concentrations were determined from the supernatant using the Bradford assay (Bio-Rad, Hercules, CA, USA, ref 5000001). Thereafter, 50 µg of proteins were then diluted in 2× Laemmli sample buffer, resolved by SDS–PAGE, and transferred onto nitrocellulose membranes (Amersham Protran Premium 0.2 µm, ref 16000004, Cytiva, Marlborough, MA). The membranes were incubated overnight at 4°C with the following primary antibodies: rabbit polyclonal phosphorylated nuclear factor-$\kappa$B (NF-$\kappa$B) p65 (1:1000, cat. no. 3033, Cell Signaling Technology, Danvers, MA, USA), monoclonal rat interleukin (IL)-10 (1:500, sc-73309, Santa Cruz Biotechnology, Dallas, TX, USA), mouse monoclonal HSP70 (1:750, ADI-SPA-810, Enzo, Farmingdale, NY, USA), mouse monoclonal HSP27 (1:750, ADI-SPA-800, Enzo Life Sciences, Farmingdale, NY, USA), rabbit polyclonal phosphorylated mammalian target of rapamycin (mTOR; 1:1000, cat. no. 2971, Cell Signaling Technology), rabbit polyclonal phosphorylated S6 ribosomal protein (1:1000, cat. no. 5364, Cell Signaling Technology), mouse monoclonal p-p70 S6 kinase (1:1000, cat. no. 9206, Cell Signaling Technology) rabbit polyclonal vascular endothelial growth factor (VEGF; 1:500, sc507, Santa Cruz Biotechnology) and rabbit polyclonal transforming growth factor $\beta$ (TGF-$\beta$ 1:1000, cat. no. 3711, Cell Signaling Technology, Danvers, MA, USA). Normalisation of total and phospho-protein was performed using Ponceau S staining. Detection was achieved using horseradish peroxidase (HRP)-linked secondary antibodies (cat. no. 7074 for rabbit, cat. no. 7076 for mouse and cat. no. 7077 for rat, Cell Signaling Technology, 1:2000, Danvers, MA, USA). Immunoblots were revealed using a Femto West HRP substrate kit (GenTex, ref GTX14698, Zeeland, MI, USA) and proteins were visualised by enhanced chemiluminescence and quantified with Image Lab software (Version 5.2.1., Bio-Rad).

**Haematoxylin and eosin.** Serial transverse sections (10 µm thick) from liquid nitrogen-cooled isopentane muscle samples embedded in optimal cutting temperature (OCT) medium (Thermo Fisher Scientific, Waltham, MA, USA) were obtained using a cryostat at −25°C

and mounted on glass microscope slides. Sections were stained with haematoxylin–eosin. Whole histological sections were imaged with a Nanozoomer 2 device (Hamamatsu Photonics, Shizuoka, Japan) with a ×20 objective in the MRI-INM platform of the Institute of Neuroscience of Montpellier (France), and analysed with NDP.view2 Image viewing software.

**Immunohistochemistry.** Serial transverse sections (10 μm thick) from liquid nitrogen-cooled isopentane muscle samples embedded in OCT medium were obtained using a cryostat at –25°C and mounted on glass microscope slides. Sections were permeabilised for 30 min in phosphate-buffered saline (PBS), 5% goat serum, 0.1% Triton X-100 at 37°C, and incubated with the primary antibody in a solution of PBS–5% goat serum for 24 h at 4°C. Sections were washed in PBS 3 × 5 min and incubated with the secondary antibody in PBS for 1 h at 37°C. Sections were washed in PBS, 3 × 5 min, incubated 30 s with Hoechst and washed once in PBS 1 min and mounted. Whole histological sections were imaged with an Axioscan device (Zeiss, Oberkochen, Germany) in the MRI-INM platform of the Institute of Neuroscience of Montpellier(France) with a ×20 objective and analysed with Zen software (v3.8.2). Primary antibodies used were anti-Lamininin (1:500, rabbit polyclonal, L9393, Sigma, Saint-Louis, MO, USA) and anti-Dystrophin (1:500, mouse monoclonal, Mandra, Developmental Studies Hybridoma Bank, Iowa City, IA, USA).

### Statistical analyses

Individual data are presented along with means ± standard deviation (SD) or median (only for muscle proteins). Normality and homoscedasticity of the results were visually assessed by inspecting Q-Q plot of the residuals. Linear mixed models (LMM) fit by the restricted maximum likelihood (REML) method were used to analyse the normality of the data distribution using treatments (CWI, TWI, HWI) and time (PRE–D10) as fixed effects and participants as random effect. A marginal model was used when a mixed model failed to compute the results of the model. Bonferroni's and Šidák's *post hoc* tests were performed when main effects were statistically significant to define the exact differences. The difference of the means, 95% confidence intervals (95% CI) of the differences between groups or between times, and Hedges' $g$ effect sizes (ES) are included alongside the *P*-values calculated using LMM. The interpretation of the Hedges' $g$ effect sizes is as follows: small effect = 0.2, medium effect = 0.5, large effect = 0.8 (Brydges, 2019). Protein results expressed a non-normal distribution and were analysed using non-parametric tests. Each treatment

group was tested separately for potential time effect with Friedman's test for repeated measures and when significant, Wilcoxon's signed ranks test was used to identify differences between times in each group. The Kruskal–Wallis test was used to identify group effects at D5 and D11 and when significant, the Mann–Whitney test was used to identify differences between treatments. A statistically significant difference was determined at $P < 0.05$. Data were analysed using SPSS Statistics 29.0.1 (IBM Corp., Armonk, NY, USA) and graphs were performed using R (version 4.4.0; R Foundation for Statistical Computing, Vienna, Austria) and Rstudio (version 2024.04.1+748) using the package ggplot2.

## Results

### Muscle damage

Total work (J/kg) performed during the muscle damage exercise was not different between groups ($P = 0.843$), independently of set number ($P = 0.735$) (Table 1). As shown in Fig. 2, myofibres infiltrated by nuclei and negative to dystrophin staining became visible following the simulated injury, thus confirming myofibres undergo cellular necrosis.

### Core body and muscle temperatures

The treatment significantly impacted core temperature (CWI: 36.9 ± 0.3°C, TWI: 37.0 ± 0.2°C, HWI: 38.0 ± 0.4°C, $P < 0.001$) with higher core temperatures for HWI than CWI (+1.1°C, 95% CI [+0.8, +1. 5]°C, $P = 0.001$) and TWI (+1.0°C, 95% CI [+0.7, +1.3]°C, $P = 0.001$) but with no significant difference between CWI and TWI (+0.1°C, 95% CI [−0.2, +0.4]°C, $P = 0.444$). Muscle temperatures were higher in HWI compared to TWI and in TWI compared to CWI at all depths of measurement ($P < 0.001$), and there was an interaction with depth ($P < 0.001$) as the difference between CWI and HWI was larger superficially (1 cm: +12.2°C) compared to the deeper measurement (3 cm: +5.8°C, Fig. 3).

### Muscle force and pain

MVC was significantly different over time with values at D2, D4 and D10 lower than PRE (e.g. MVC: D2: −52 kg, 95% CI [−60, −45] kg, D4: −37 kg, 95% CI [−45, −30] kg, D10: −29 kg, 95% CI [−37, −21] kg, $P < 0.001$, ES ≥ 1.16, Fig. 4); no statistically significant treatment effects ($P = 0.685$) nor interactions ($P = 0.108$, ES ≤ 0.74) were identified.

Muscle pain measured both by self-palpation and following squats was significantly different over time ($P < 0.001$) with an initial increase up to D2 (ES ≥ 2.89), a

decrease from D3 to D6, and no difference with PRE from D7 (Fig. 5). Muscle pain measured by self-palpation also differed between treatments ($P = 0.0350$, without interaction $P = 0.988$), with lower pain in HWI than TWI ($-1.8$ a.u, 95% CI [$-3.4$, $-0.1$] a.u., $P = 0.0346$, ES $\geq -0.87$), while CWI did not differ from other conditions ($P \geq 0.207$, ES $\geq -0.64$). Muscle pain measured during a squat exercise remained similar across treatments ($P = 0.364$, without interaction $P = 0.966$, ES $\geq -0.70$).

### Blood markers

There was an interaction between the effect of time and treatment for both CK and myoglobin ($P \leq 0.022$, Fig. 6). Of note, because 32 participants exhibited CK values on D4 that were higher than the assay detection limits, D4 data for CK were not included in statistical analysis. CK levels were higher at D8 than baseline for all treatments ($P \leq 0.0171$) and were lower in HWI than TWI ($-2163$ U/L, 95% CI [$-3949$, $-377$] U/L, $P < 0.0123$, ES $= -1.12$) and CWI ($-2700$ U/L, 95% CI [$-4407$, $-994$] U/L, $P < 0.001$, ES $= -1.14$), without differences between CWI and TWI ($-537$ UL, 95% CI [$-1212$, $2287$] U/L, $P = 1.000$, ES $= 0.19$). Myoglobin levels increased at D4 for all treatments ($P \leq 0.001$) and returned to baseline by D8 ($P = 1.000$), and HWI was lower at D4 than both TWI ($-897$ μg/l, 95% CI [$-1451$, $-342$] μg/l, $P < 0.001$, ES $= -1.16$) and CWI

($-1073$ μg/l, 95% CI [$-1616$, $-531$] μg/l, $P < 0.001$, ES $= -1.15$), without any differences between CWI and TWI ($176$ μg/l, 95% CI [$-365$, $719$] μg/l, $P = 1.000$, ES $= 0.17$).

### Protein expression of heat shock proteins

Non-parametric analyses showed that HSP27 increased over time in HWI ($P = 0.0273$) but not in CWI or TWI ($P \geq 0.121$, Fig. 7). HSP27 expression at D11 differed between treatments ($P = 0.0465$) with higher expression in HWI than CWI ($P = 0.00762$). HSP70 increased in both TWI ($P = 0.0445$) and HWI ($P < 0.0379$) but not CWI ($P = 0.670$, Fig. 7); however, there was no significant difference between treatments at D5 ($P = 0.779$). While the overall difference between treatments at D11 was only $P = 0.0532$, HWI was significantly higher than CWI ($P = 0.0294$).

### Protein expression of p-NF-$\kappa$B and IL-10

Phospho-NF-$\kappa$B (p-NF-$\kappa$B) expression levels increased in both TWI and CWI ($P \leq 0.00584$) but remained stable in HWI ($P = 0.497$, Fig. 8). In particular, p-NF-$\kappa$B differed between treatments at D11 ($P = 0.00821$) with higher expression in both CWI ($4.06 \pm 3.57$) and TWI ($8.47 \pm 8.42$) than HWI ($1.49 \pm 2.34$; $P \leq 0.0196$).

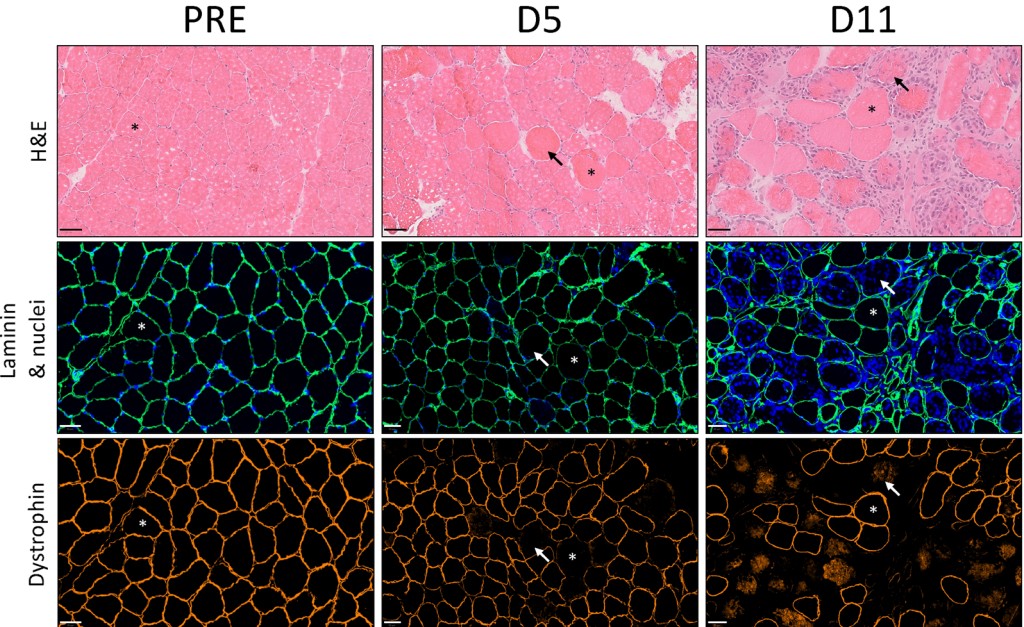

**Figure 2. Microscopy images of histological staining illustrating the effect of the muscle damage protocol at D5 and D11 in comparison to PRE**
H&E, laminin (green), nuclei (blue) and dystrophin (orange) staining are displayed. The asterisk identifies the same fibre across the different stains for each sampling time. Examples of damaged fibres are identified by arrows at D5 and D11; these fibres present a swollen aspect with a loss of normal polygonal outline, mononuclear cell infiltration, internal nuclei or a loss of dystrophin immunoreactivity of their membrane. A large number of similar fibres presenting necrosis are visible in the D11 illustration. Scale bars, 50 μm.

IL-10 expression remained stable in CWI ($P = 0.0662$) and TWI ($P = 0.416$) but increased in HWI ($P = 0.0135$, Fig. 8). IL-10 levels differed between treatments at D11 ($P = 0.0453$) with higher expression in HWI than CWI ($2.13 \pm 5.04$ *vs*. $0.90 \pm 0.94$, $P = 0.0135$).

### Protein expression of protein synthesis pathway

p-mTOR expression remained stable in CWI ($P = 0.717$) but increased in both TWI ($P = 0.00178$) and HWI ($P = 0.0450$, Fig. 9) without reaching statistically significant differences between treatments at D5 ($P = 0.101$) or D11 ($P = 0.163$). p-S6 expression increased in all treatments ($P \leq 0.00248$, Fig. 9) but was not different between treatments at D5 ($P = 0.300$) and D11 ($P = 0.0811$). p-p70-S6K remained stable over time in all treatments ($P \geq 0.156$, Fig. 9).

### Protein expression of VEGF and TGF-$\beta$1

VEGF expression (Fig. 10) remained stable in CWI ($P = 0.529$) but decreased in both TWI ($P = 0.00150$) and HWI ($P = 0.0202$). A significant treatment effect appeared at D11 ($P = 0.0105$), with higher expression in CWI than TWI (CWI: $0.74 \pm 0.71$ *vs*. TWI: $0.45 \pm 0.36$, $P = 0.00565$) and HWI (CWI: $0.74 \pm 0.71$ *vs*. HWI: $0.56 \pm 0.19$, $P = 0.0241$).

TGF-$\beta$1 (Fig. 11) remained stable over time in all treatments ($P \geq 0.223$).

## Discussion

The present study aimed to investigate the effects of thermal interventions in a post-injury muscle regeneration model following the induction of large muscle damage and myofibre necrosis. Contrary to our hypothesis, CWI treatment did not alleviate muscle pain. Conversely, data showed that HWI exposure limits pain and favours muscle regeneration as indicated by lower circulating markers of muscle damage (both CK and myoglobin), lower p-NF-$\kappa$B activity, and an upregulation of IL-10 expression compared to CWI and TWI treatments. This is the first study to examine the effects of CWI and HWI on muscle regeneration following an electrically stimulated eccentric exercise.

### Functional outcomes

The muscle damage protocol induced a significant decrease in MVC from PRE to D2 ($-59\%$, 95% CI [$-68\%$, $-51\%$]), which was not fully recovered by D10 ($-32\%$, 95% CI [$-38\%$, $-27\%$]). It is worth noting that the reduction in force capacity appeared larger than in previous studies using electrically stimulated contractions,

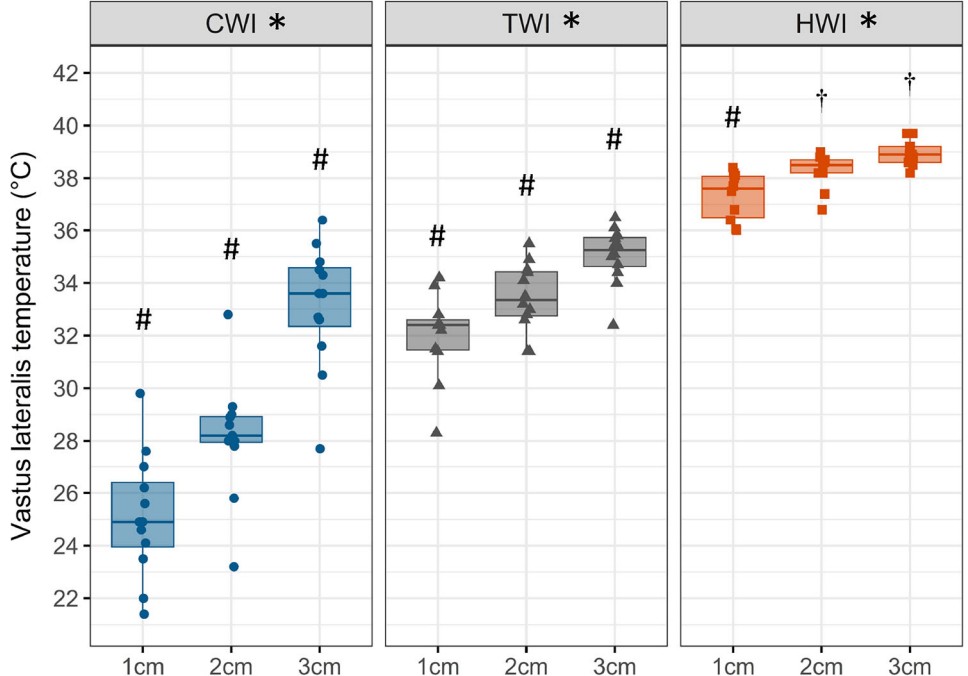

**Figure 3. Vastus lateralis muscle temperature at 1, 2 and 3 cm depth from skin surface after CWI, TWI and HWI interventions**
Data are presented as boxplot with median and interquartile range along with individual data. *Significantly different between group ($P < 0.05$); #significantly different from all depths in the group ($P < 0.05$); †significantly different from 1 cm in the same group ($P < 0.05$).

potentially due to the use of a higher electrical stimulation intensity during the muscle damage protocol (Crameri et al., 2007; Mackey et al., 2016; Nosaka et al., 2011). The current data did not show a significant effect of the thermal interventions on force recovery (Fig. 4). This aligns with a part of the literature showing no benefit of CWI on isometric force recovery following exercise-induced muscle damage or resistance exercise (Crystal et al., 2013; Glasgow et al., 2014; Goodall & Howatson, 2008; Howatson et al., 2009; Machado et al., 2017; Sautillet et al., 2024; Vieira Ramos et al., 2016). Interestingly, several studies reported a beneficial effect of CWI on isometric peak torque, squat jump and counter movement jump performance recovery; however, the exact mechanisms remain unclear (Chaillou et al., 2022; Leeder et al., 2012; Pointon et al., 2012; Roberts, Muthalib, Stanley, et al., 2015; Vaile et al., 2008). In contrast, chronic CWI exposure for 12 weeks following resistance training was shown to lower force and muscle mass adaptations elicited by resistance training (Roberts, Raastad, Markworth, et al., 2015). If our results reported neither benefit nor detrimental effect of a relatively short-term CWI on force recovery, based on the existing literature, it could be hypothesised that chronic CWI used post-injury might interfere with rehabilitation processes by reducing force recovery and attenuating hypertrophy signalling.

Furthermore, cryotherapy's most intended purpose is probably a reduction in muscle pain and soreness following injury or muscle damage (Frery et al., 2023; Leeder et al., 2012). Indeed, icing of the ankle in the absence of prior pain has been shown to increase pain threshold and pain tolerance by reducing nerve conduction velocity (Algafly et al., 2007). However,

equivocal evidence regarding the effect of CWI on muscle pain and soreness following exercise-induced muscle damage has been reported. A part of the literature reports reductions in muscle pain and soreness (Hohenauer et al., 2020; Huang et al., 2024; Leeder et al., 2012; Machado et al., 2016; Moore et al., 2022; Siqueira et al., 2018), while in contrast, several studies report no significant benefit or detrimental effects of CWI (Glasgow et al., 2014; Goodall & Howatson, 2008; Howatson et al., 2009; Sautillet et al., 2024; Sellwood et al., 2007; Vaile et al., 2008). Large methodological differences in the muscle damage/injury models, cooling modalities and pain assessment could explain these discrepancies. To our knowledge, the only study that examined the effect of cryotherapy on muscle tear injury showed no benefit of chronic icing (20- to 30-min applications) during the first 36 h post-injury on muscle pain perception at rest or during physical activity (Prins et al., 2011). Accordingly, the current data showed no benefit of CWI for pain perception following a simulated injury (Fig. 5).

Conversely, while the interaction did not reach significant values ($P = 0.108$), the decrease in MVC at D4 was clinically lower in HWI than other groups (HWI: −31% [−40%, −22%] *vs.* CWI and TWI: −47%

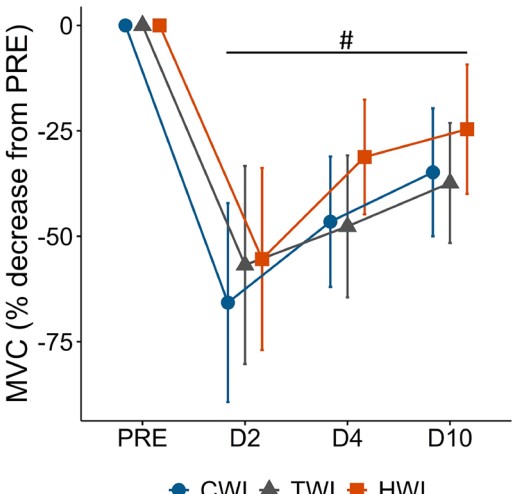

**Figure 4. Maximal voluntary contraction force**
Data are presented as means and standard deviation of the percentage decrease of MVC from PRE. #Significant time effect (*P* < 0.05).

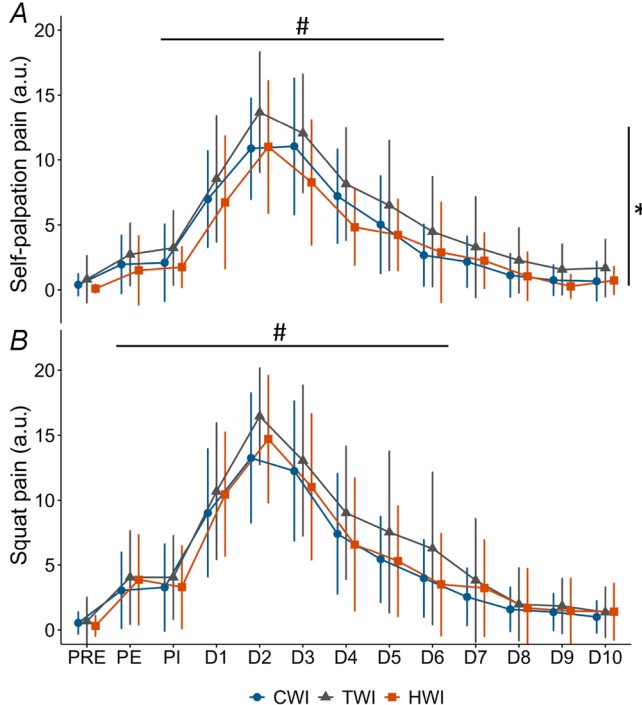

**Figure 5. Muscle pain measured during self-palpation (*A*) and after three squats (*B*) using an arbitrary unit scale (0–20) where 0 represented no pain and 20 extremely painful**
Data are presented as means and standard deviation of the perceived pain score. PE: post-exercise, PI: post water immersion. #Significant time effect showing differences from PRE (*P* < 0.05); *significant group effect between HWI and TWI (*P* < 0.05).

[−57%, −38%], ES ≥ 0.61, Fig. 4). Furthermore, self-palpation muscle pain was significantly lower in HWI than TWI (Fig. 5). These results extend recent observations that a single CWI did not improve the loss in rate of force development or pressure pain threshold following exercise-induced muscle damage, whereas HWI mitigated both (Sautillet et al., 2024). Those results also broaden the beneficial effects of local heating previously reported on muscle pain and soreness following eccentric exercise (Mayer et al., 2006; Petrofsky et al.,

2017; Sautillet et al., 2023, 2024; Wang et al., 2022). Although, the mechanisms promoting improvements in force recovery and pain perception using HWI remain unclear, several explanations have been proposed. As HWI increases blood flow and vasodilatation (Cheng et al., 2021; Francisco et al., 2021; Heinonen et al., 2011), it has been suggested to promote healing by increasing nutrient and oxygen supply to the injury and elicit hypoalgesia by accelerating the removal of factors sensitising muscle nociceptors (Kim et al., 2019; Malanga et al.,

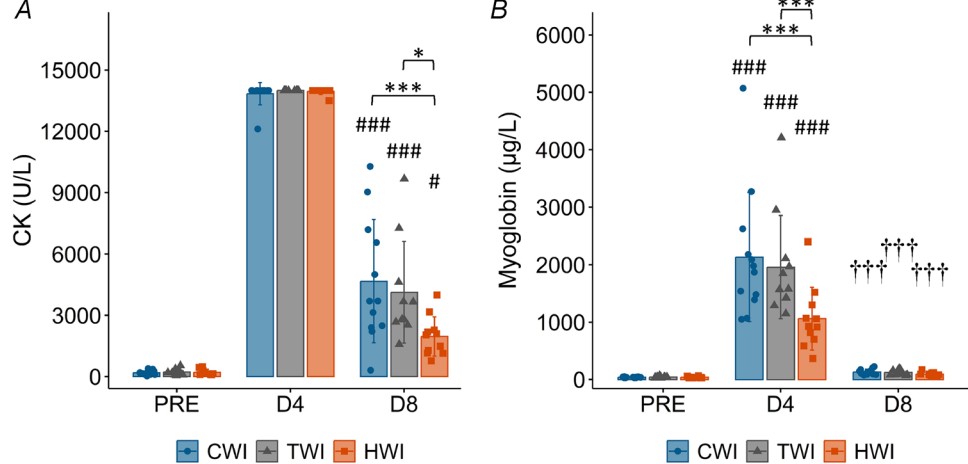

**Figure 6. The blood markers creatine kinase (CK) (*A*) and myoglobin (*B*)**
Data are presented as means and standard deviation along with individual data. CK at D4 is displayed in the graph but data were excluded from statistical analysis. #Significantly different from PRE ($P < 0.05$); ###significantly different from PRE ($P < 0.001$); †significantly different from D4 ($P < 0.05$); †††significantly different from D4 ($P < 0.001$); *significantly different between groups ($P < 0.05$); ***significantly different between groups ($P < 0.001$).

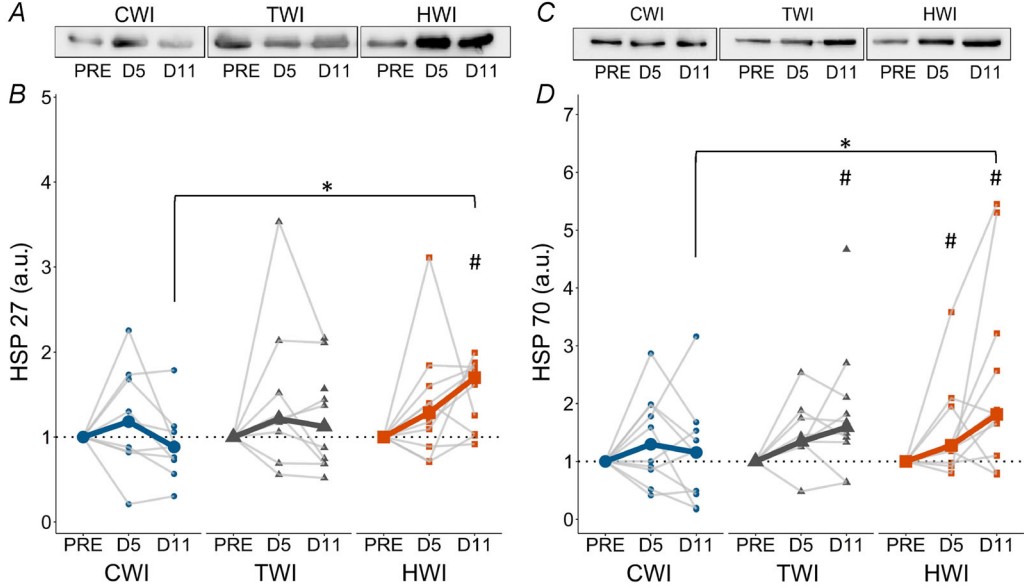

**Figure 7. Representative blots and quantification of HSP 27 (*A*, *B*) and HSP 70 (*C*, *D*) protein expression at PRE, D5 and D11 in CWI, TWI and HWI groups**
Data are presented as medians along with individual data. #Significantly different from PRE ($P < 0.05$); *significantly different between groups ($P < 0.05$).

2015). Indeed, mechanical hyperalgesia following an eccentric muscle damage has been partly linked to two neurotrophic factors, nerve growth factor (NGF) and glial cell line-derived neurotrophic factor, that may stimulate muscle nociceptors (Mizumura & Taguchi, 2016; Peake, Neubauer, Della Gatta, et al., 2017). Interestingly, in rats, heat exposure using hot packs has been shown to reduce NGF expression levels and decrease physical inactivity-induced mechanical hyperalgesia (Nakagawa et al., 2018). In contrast, no significant modulation of NGF was found after a CWI following intense resistance exercise in humans (Peake, Roberts, Figueiredo, et al., 2017). Although neurotrophic factors were not measured in the current study, blood sample results indicate a potential acceleration in the removal of markers of muscle damage, also possibly related to the observed hypoalgesia.

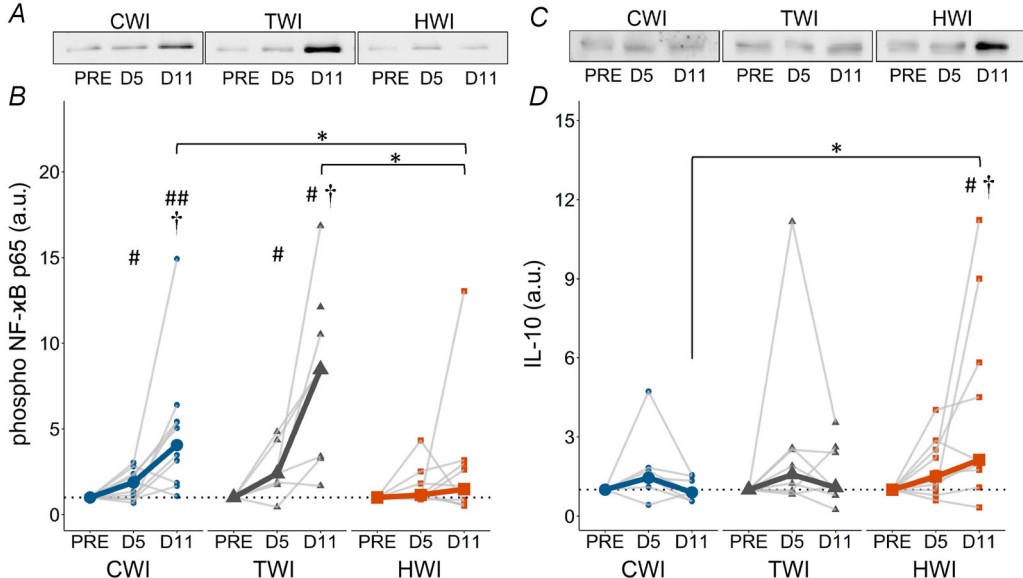

**Figure 8. Representative blots and quantification of phosphorylated NF-κB p65 (*A*, *B*) and IL-10 (*C*, *D*) protein expression at PRE, D5 and D11 in CWI, TWI and HWI groups**
Data are presented as medians along with individual data. #Significantly different from PRE ($P < 0.05$); ##significantly different from PRE ($P < 0.01$); †significantly different from D5 ($P < 0.05$); *significantly different between groups ($P < 0.05$).

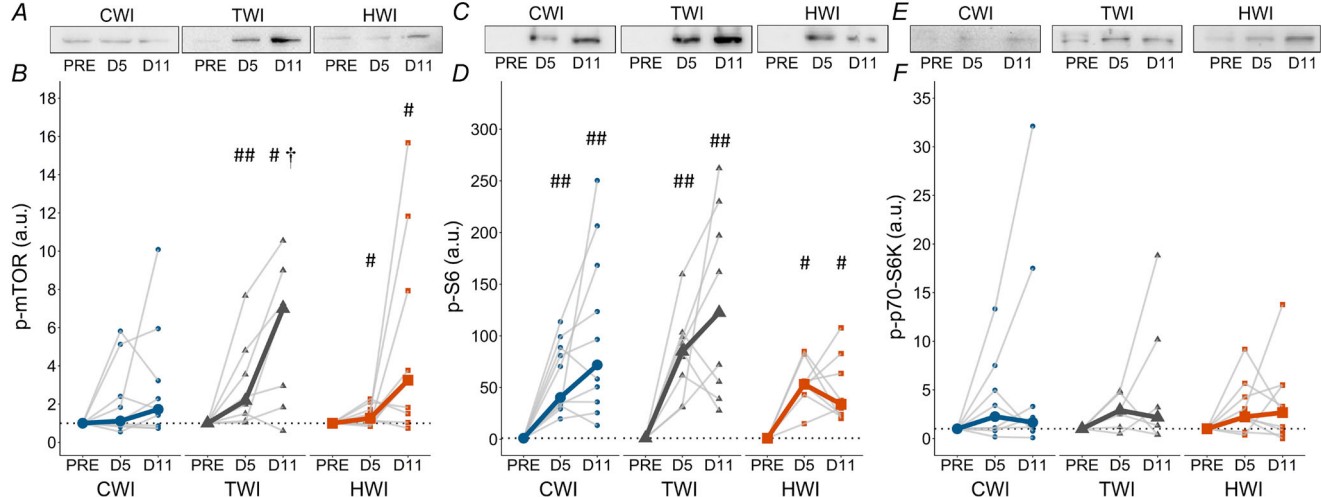

**Figure 9. Representative blots and quantification of phosphorylated mTOR (*A*, *B*), phosphorylated-S6 (*C*, *D*) and phosphorylated-p70 (*E*, *F*) protein expression at PRE, D5 and D11 in CWI, TWI and HWI groups**
Data are presented as medians along with individual data. #Significantly different from PRE ($P < 0.05$); ##significantly different from PRE ($P < 0.01$); †significantly different from D5 ($P < 0.05$); *significantly different between groups ($P < 0.05$).

## Markers of muscle damage

As per previous studies (Karlsen et al., 2020; Mackey et al., 2016), the current muscle damage protocol induced a large and sustained increase in circulating markers of muscle damage (Fig. 6). The literature on the effect of CWI on CK and myoglobin levels following eccentric or resistance exercise remains equivocal with various studies showing a decrease (Huang et al., 2024; Leeder et al., 2012; Roberts et al., 2014; Siqueira et al., 2018; Vaile et al., 2008; Vieira et al., 2016), no significant effect (Ahokas et al., 2020; Glasgow et al., 2014; Goodall & Howatson, 2008; Howatson et al., 2005, 2009; Machado et al., 2016; Peake, Roberts, Figueiredo, et al., 2017; Sellwood et al., 2007), or an increase (Tseng et al., 2013). In the context of muscle injury, one of the suggested benefits of CWI is to minimise secondary cell damage (Bleakley & Davison, 2010b; Merrick, 2002). However, our results show no significant effect of CWI on circulating markers of muscle damage (Fig. 6) suggesting no benefit of CWI in limiting secondary injury or improving muscle recovery from large muscle damage. Conversely, our results showed reduced levels of CK and myoglobin following HWI, suggesting that HWI potentially increased the rate at which CK/myoglobin was removed from the muscle and entered the bloodstream. Indeed, as illustrated previously in the literature, the increase in blood flow and the vasodilatation elicited by the HWI are likely to have played a large role in the observed reduction in circulating markers of muscle damage (Cheng et al., 2021; Francisco et al., 2021; Heinonen et al., 2011).

## Muscle regeneration

While the precise role of HSPs in post-injury recovery remains unclear, the literature indicates that HSPs may have played a key role in protecting and facilitating skeletal muscle regeneration. HSPs have been suggested to improve cell survival and facilitate muscle recovery (Fennel et al., 2022; Paulsen et al., 2007; Thompson et al., 2001). Indeed, HSP27 has been shown to bind to cytoskeletal/myofibrillar proteins immediately following an eccentric exercise and potentially stabilise the disrupted myofibrillar structures (Paulsen et al., 2007). In injured rats, HSP72 expression has been reported to increase following heat stress treatment (Kojima et al., 2007; Oishi et al., 2009; Shibaguchi et al., 2016) but not after local icing (20 min at 0°C) (Shibaguchi et al., 2016). According to the literature, the observed HSP27 and HSP70 upregulations in HWI could have been partly responsible for the reduced CK/myoglobin levels through a reduction in the extent of muscle damage (Fennel et al., 2022). However, the reduction in CK/myoglobin levels appears earlier than the observed upregulation in HSPs. Of note, as no early muscle sample (D0 to D5) was collected, the complete kinetics of HSP remains unknown in this study. Furthermore, in animals, HSP72 appears

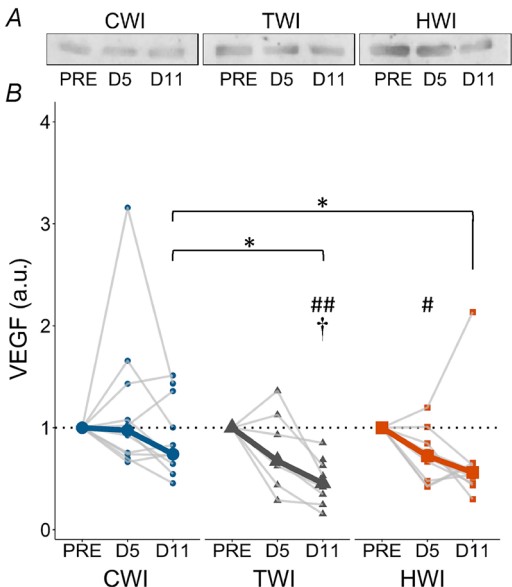

**Figure 10. Vascular endothelial growth factor (VEGF) representative blots (*A*) and quantification (*B*) of protein expression at PRE, D5 and D11 in CWI, TWI and HWI groups**
Data are presented as median along with individual data. #Significantly different from PRE ($P < 0.05$); ##significantly different from PRE ($P < 0.01$); †significantly different from D5 ($P < 0.05$); *significantly different between groups ($P < 0.05$).

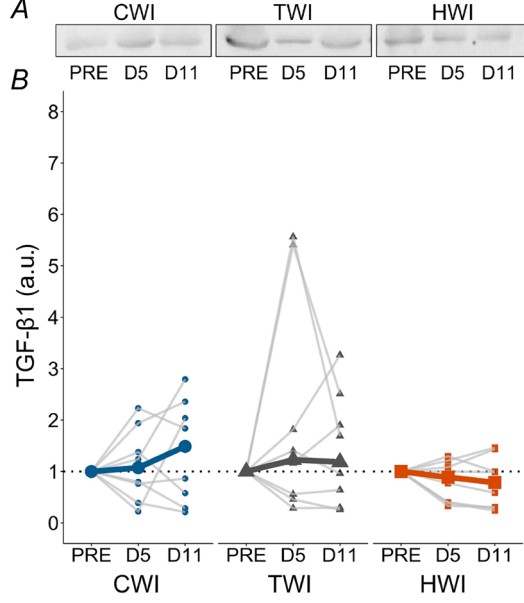

**Figure 11. Transforming growth factor-$\beta$1 (TGF-$\beta$1) representative blots (*A*) and quantification (*B*) of protein expression at PRE, D5 and D11 in CWI, TWI and HWI groups**
Data are presented as median along with individual data.

to play a crucial role in preventing activation of NF-$\kappa$B (Shi et al., 2006; Wong et al., 1999), which is a major transcription factor inducing skeletal muscle atrophy, favouring pro-inflammatory mechanisms and inhibiting myogenesis (Bakkar & Guttridge, 2010; Mourkioti & Rosenthal, 2008). Importantly, heat stress has been shown to upregulate HSP70/72 expression and downregulate TNF-$\alpha$/NF-$\kappa$B pathway in rats (Ohno et al., 2010; Senf et al., 2008, 2013), showing a negative correlation between HSP72 and NF-$\kappa$B (Ohno et al., 2010). Similarly, the current data showed an earlier increase in HSP70 in HWI and no significant increase of p-NF-$\kappa$B (Figs 7 and 8), potentially confirming the preventive role of HSP70 against NF-$\kappa$B in humans. Of note, p-NF-$\kappa$B was significantly upregulated at all times post-injury in the TWI group despite a significant increase in HSP70 at D11. Thus, additional investigations are necessary to further understand the effect of HSPs on muscle regeneration.

It appears important to note that a muscle temperature/stimulus threshold might be necessary to elicit an upregulation in HSPs. Indeed, contrary to our results, two studies in humans found no effect of muscle heating on muscle HSP expression, one using 5 days of 90 min with a heating garment (54°C–55°C perfused water) and the other using a single 20-min water immersion (46°C) (Fuchs, Smeets, Senden, et al., 2020; Kim et al., 2019). Relatively low increases in muscle and/or skin temperatures were reported, below the temperature previously reported to induce an HSP upregulation (Fennel et al., 2022; Hafen et al., 2019). These studies also confirm that all heating modalities are not suitable to induce high muscle temperatures, as, for example, a water-perfused suit was shown to induce insufficient heat stress to impact signalling pathways (Ihsan et al., 2020). Furthermore, recent studies using HWI following muscle damage (Sautillet et al., 2023, 2024) hypothesised that the beneficial effects reported for HWI were mostly due to a core temperature increase above 38.5°C. However, the results of our study contradict this theory. If high core temperatures might elicit additional benefits, in our study, the large use of cooling solutions on the upper body limited core temperature to 38°C at the end of HWI. Our results and the literature suggest that a muscle temperature increase, rather than a core temperature increase, is necessary to induce muscle HSP upregulation (Hafen et al., 2019).

NF-$\kappa$B expression has been reported to rapidly rise within the first hours following exercise-induced muscle damage (Hyldahl et al., 2011; Jameson et al., 2021) and closely regulate TNF-$\alpha$, a pro-inflammatory cytokine that is also a potent activator of NF-$\kappa$B, forming a positive feedback loop for NF-$\kappa$B (Bakkar & Guttridge, 2010; Li et al., 2008). In animal studies, post-injury cryotherapy has been shown to decrease NF-$\kappa$B and TNF-$\alpha$ expression (Dos Santos Haupenthal et al., 2021; Kawashima et al.,

2021; Miyazaki et al., 2023; Vieira Ramos et al., 2016). In contrast, the current data obtained in humans showed an increase in p-NF-$\kappa$B expression at D11 in both TWI and CWI, suggesting that CWI had no significant effect on the NF-$\kappa$B/TNF-$\alpha$ pathway during *in vivo* human muscle regeneration. In contrast, HWI blunted p-NF-$\kappa$B expression suggesting a potential attenuation in pro-inflammatory and pro-atrophy pathways between D5 and D11. The previously highlighted increase in HSP70 is likely to have played a role in p-NF-$\kappa$B reduction (Ohno et al., 2010; Senf et al., 2008, 2013). Moreover, in cultured cells, heat stress was shown to increase expression of I$\kappa$B$\alpha$, an inhibitor of NF-$\kappa$B (Ohno et al., 2011; Wong et al., 1999), while simultaneously decreasing the expression of p-NF-$\kappa$B and increasing protein synthesis (Ohno et al., 2011). Although we did not measure I$\kappa$B$\alpha$ expression, our findings also suggest a heat-related inhibition of NF-$\kappa$B. Furthermore, full body heat exposure during leg immobilisation has recently been shown to decrease the ratio of phosphorylated NF-$\kappa$B, confirming the impact of heat on the NF-$\kappa$B pathway (Labidi et al., 2024).

Anti-inflammatory processes are largely affected by IL-10 expression, which is particularly upregulated during the late inflammatory phase (Deng et al., 2012; Tidball, 2017). Kawashima et al. (2021) reported that an icing treatment post-injury led to lower expression of IL-10 that coincided with a reduced number of M2 macrophages, thereby delaying the regeneration (Kawashima et al., 2021). In contrast, while no specific study has examined the effect of heat on IL-10 and M2 macrophage phenotype, heat appeared to shorten the inflammatory phase by hastening the peak of macrophages appearance (Takeuchi et al., 2014). Our results on p-NF-$\kappa$B and IL-10 suggest that the differences in inflammatory kinetics observed in animals between the different thermal interventions could also be present in humans (Fig. 8). However, further investigations are necessary to confirm the effect of HWI on the inflammatory shift. Although CWI treatment differed from HWI in terms of NF-$\kappa$B and IL-10 expression, CWI was not different from TWI. Similar to previous results showing no effect of CWI on inflammation following a bout of resistance, our results suggest that the CWI protocol used in this study had no effect on the inflammatory markers measured following a simulated injury (Peake, Roberts, Figueiredo, et al., 2017).

Angiogenesis is a key process of muscle regeneration, necessary to restore the integrity of muscular and vascular tissue following injury (Huey et al., 2016; Miyazaki et al., 2011; Ochoa et al., 2007). In a study investigating the effect of icing on post-injury angiogenesis in rats, a lower expression of VEGF was observed 3 days post-injury concomitantly with smaller vessel volumes (Singh et al., 2017). Although it has never been investigated during large muscle damage recovery in humans, a chronic CWI post-resistance training of 12 weeks was reported

to maintain VEGF gene expression and increase the number of capillaries per fibre compared to an active recovery (D'Souza et al., 2018). Similarly to the previous results on VEGF in humans, no significant change in the time course of VEGF response was observed in the CWI group (Fig. 10). In contrast, VEGF levels were significantly lower in TWI and HWI at D11, suggesting a maintenance of pro-angiogenic factors following CWI but not following HWI. Interestingly, in humans, a 90-min heating garment applied after an eccentric exercise increased VEGF expression after 1 day, although similarly to our results, this effect was not sustained at 5 days (Kim et al., 2019). However, the biopsy timing (D5 and D11) did not allow us to detect a possible early change in VEGF expression. Future studies should more closely examine the kinetics of VEGF activity and its consequences in the muscle regeneration process.

While no human studies have investigated mTOR regulation following cryotherapy application on injury, studies in injured-animal models suggest that cooling downregulates insulin-like growth factor 1 (IGF-1), the upstream regulator of mTOR, potentially reducing protein synthesis (Kawashima et al., 2021; Takagi et al., 2011). Furthermore, post-resistance exercise CWI has been found to decrease satellite cell recruitment, ribosome biogenesis and p70S6K activation (Roberts, Raastad, Markworth, et al., 2015), while no significant difference in the IGF-1 signalling pathway was reported compared to an active control group (Peake et al., 2020). Interestingly, in humans, CWI was reported to reduce post-exercise myofibrillar protein synthesis rate without apparent reduction in signalling proteins that regulate translation initiation (i.e. mTOR, p70S6KThr421/Ser424, rpS6 and 4E-BP1) (Fuchs, Kouw, Churchward-Venne, et al., 2020). Although heat stress has been shown to upregulate the AKT/mTOR pathway and its downstream effectors in animal models compared to a control group (Ohno et al., 2012; Uehara et al., 2004; Yoshihara et al., 2013), our findings in humans do not confirm these previous results. However, HWI did not appear to downregulate mTOR signalling, while CWI, which partially impaired muscle regeneration efficiency, showed no significant upregulation of p-mTOR, potentially suggesting a reduced protein synthesis (Fig. 9). Similarly to VEGF, the timing of biopsies may explain the lack of p70-S6K phosphorylation observed.

Despite the necessary role of TGF-$\beta$1 during muscle regeneration, excessive elevations have been associated with fibrosis development (Delaney et al., 2017). In rats, excessive collagen formation was reported following a single post-injury ice application (Shibaguchi et al., 2016; Takagi et al., 2011) in contrast to heat application, which limited fibrosis development (Shibaguchi et al., 2016; Takeuchi et al., 2014). Contrary to the literature in animals, we did not find any differences in TGF-$\beta$1 expression between treatments; however, a larger variability was observed for CWI and TWI compared to HWI (Fig. 11). Although the previous data presented for p-NF-$\kappa$B and IL-10 expression suggest differences in inflammatory activity in HWI only, TGF-$\beta$1 remained weakly expressed in HWI.

## Methodological considerations, limits and perspective

Consistent with previous reports (Højfeldt et al., 2023; Mackey & Kjaer, 2017), no participant fully recovered strength by the end of the trial. Based on the current results, future studies should examine the effect of thermal interventions for periods longer than 2 weeks.

As highlighted by Kim et al. (2019) as a limitation in their study, the benefits of cryotherapy and heat therapy may be partly attributable to a placebo effect (Kim et al., 2019; Wilson et al., 2018, 2019). Although blinding the thermal interventions was not possible, participants remained unaware of the study's hypothesis. Moreover, we noticed a strong positive belief towards the efficacy of CWI among participants, suggesting that the benefits of heat exposures are unlikely due to a placebo effect. Thermoneutral immersion (TWI group) was chosen as a sham modality to isolate the effects of the thermal intervention and reduce potential placebo or physiological effects from the immersion. It should be noted that the differences in immersion durations did not allow us to determine whether the effects were specific to the temperature involved or in combination with hydrostatic pressure. Although very unlikely, it is not possible to determine whether hydrostatic pressure influenced muscle recovery from the simulated injury (Versey et al., 2013; Wilcock et al., 2006). Water immersions induced larger modifications in temperature in peripheral tissue (1 cm depth) than in deep tissue (3 cm depth) (Fig. 3). However, all tissue depths were impacted by the thermal treatments, as well as central body temperature. Future studies should include a measurement of subcutaneous fat tissue and muscle thickness to improve the precision of the thermal intervention (Rodrigues et al., 2024). It should be acknowledged that the muscle temperature measurement was performed only on the unexercised limb to reduce participant discomfort in the injured leg and avoid potential interference between subsequent biopsies from the injured leg (Long et al., 2023). However, the effect of the simulated injury on the intra-muscular temperature at the end of the intervention remains unknown. Potential differences in muscle temperature between the exercised and unexercised limbs, which were not accounted for during this study, could exist. In contrast, while the single core temperature measurement performed during this study does not allow definitive conclusions, the available literature suggests that the thermal interventions were unlikely to have triggered a cold or heat acclimation

response during the experiment (Gordon et al., 2019; Périard et al., 2015).

A growing number of studies have used electrically stimulated eccentric contraction models to investigate muscle regeneration processes (Crameri et al., 2007; Karlsen et al., 2020; Mackey & Kjaer, 2017; Mackey et al., 2016). For example, Karlsen et al. (2020) previously reported a comparable proportion of regenerating fibres using a similar muscle damage protocol to ours, while observing differences in muscle soreness and CK activity between young and elderly groups (Karlsen et al., 2020). Therefore, in our study, despite tailoring the muscle damage protocol for each participant, as done by Karlsen et al. (2020), it must be acknowledged that group differences could have been affected by inter-individual variability in the responses to the injury model (Karlsen et al., 2020). The injury model used in this study induced a large myofibre necrosis, which allowed for muscle sampling and the investigation of muscle regeneration processes in a more standardised way compared to other traumatic skeletal muscle injury models (Mackey & Kjaer, 2017). Considering the differences between traumatic skeletal muscle injuries, it remains unclear how well our findings on thermal therapies apply to other types of muscle injuries (Edouard et al., 2023). Future studies should aim at examining more specifically the effect of thermal therapies in injured athletes.

Of note, this research project was open to male and female participants. However, only one female volunteered to participate and could not be included as she did not match the criteria for group counterbalancing. This is an unfortunate limitation as there are large differences in muscle regeneration recently reported between sexes (Luk et al., 2021; You et al., 2023). While this limitation could be partly related to the local culture where the study was conducted, it is consistently more difficult to include women when the study may leave a scar due to muscle biopsies. Future studies are encouraged to include more women to participate by promoting a more inclusive protocol or encouraging multi-site collaboration.

Although participants received nutritional interventions from a sport dietician and were reminded several times by the research team, their compliance with the nutritional instructions was not monitored. Furthermore, it should be noted that the current study lacks histochemical evaluations, such as macrophages or satellite cell quantifications in the injured muscle.

## Conclusion

To our knowledge, this is the first study to investigate the effect of different thermal interventions on recovery from severe muscle damage with myofibre necrosis in humans. The present data showed that CWI neither reduced perceived pain nor provided any benefits on molecular markers of muscle regeneration. In contrast, HWI showed the potential to limit muscle pain, reduce circulating markers of muscle damage (CK and myoglobin), and to influence muscle regeneration by limiting the increase in p-NF-$\kappa$B expression and upregulating the expression of IL-10, HSP27 and HSP70. Thus, despite cryotherapy being a popular intervention in treating muscle injury, our results do not support its use to improve muscle healing. In contrast, heat therapy could offer beneficial effects on muscle recovery and may represent a promising solution to accelerate muscle healing. Further studies are needed to confirm this hypothesis, identify the most effective protocols in terms of temperature and exposure duration, as well as to understand the underlying molecular mechanisms (e.g. modulation of the myogenic programme).

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

## Additional information

### Data availability statement

Data are available from the corresponding author upon reasonable request.

### Competing interests

The authors declare they have no competing interests.

### Author contributions

The experiments were performed in Aspetar Sport Medicine Hospital, Doha, Qatar. Conception and design of the study were done by V.D., S.R., H.B., R.C., A.S. and M.C. Data collection was performed by V.D., A.M., M.A.M., E.P., B.O., T.M.F., F.Z., N.N., M.A., F.B. and S.R. Analysis and interpretation of the data were performed by V.D., T.N.G., H.B. and S.R. Initial drafting of the work was performed by V.D., S.R. and H.B. and revision of the work was performed by all authors. All authors approved the final version of the manuscript and agree to be accountable for all aspects of the work in ensuring that questions related to the accuracy or integrity of any part of the work are appropriately investigated and resolved. All persons designated as authors qualify for authorship, and all those who qualify for authorship are listed.

### Funding

This work was supported by an Aspire Zone Foundation Research grant. Adele Mornas travel expenses were supported by the Sport, Exercise and Performance laboratory (SEP) of the French institute of sport (INSEP) (Paris, France).

### Acknowledgements

The authors would like to thank all the volunteers for their participation and large implication during this study. We also acknowledge Khouloud Mtibaa and Mariem Labidi for their help with participant's recruitment and Aspetar laboratory for their excellent support with blood sample collection and analysis. The authors thank Florence Sabatier for providing expert technical assistance with biochemical analysis. The authors acknoledge the Sport, Exercise and Performance laboratory (SEP) of the French institute of sport (INSEP) (Paris, France) that funded the travel expenses to Aspetar hospital of

one of our authors (AM). Furthermore, we would like to thank Christopher John Esh for his support and technical expertise during this study.

## Keywords

CWI, damage, electrically stimulated eccentric contraction, HWI, muscle injury

## Supporting information

Additional supporting information can be found online in the Supporting Information section at the end of the HTML view of the article. Supporting information files available:

**Peer Review History**

