## [Peer Review History · The Journal of Physiology]

Muscle regeneration is improved by hot water immersion but unchanged by cold following a simulated musculoskeletal injury in humans

Valentin Dablainville, Adele Mornas, Tom Normand-Gravier, Maha Al-Mulla, Emmanouil Papakostas, Bruno Olory, Theodorakys Marín Fermín, Frantzeska Zampeli, Nelda Nader, Marine Alhammoud, Freya Bayne, Anthony Sanchez, Marco Cardinale, Robin Candau, Henri Bernardi, and Sebastien Racinais

DOI: 10.1113/JP287777

Corresponding author(s): Sebastien Racinais (sebastien.racinais@creps-montpellier.sports.gouv.fr)

The following individual(s) involved in review of this submission have agreed to reveal their identity: Geoffrey Minett (Referee #1); Jonathan M Peake (Referee #2)

Review Timeline:

Submission Date:	30-Sep-2024
Editorial Decision:	08-Nov-2024
Revision Received:	27-Jan-2025
Editorial Decision:	27-Feb-2025
Revision Received:	19-Mar-2025
Editorial Decision:	15-Apr-2025
Revision Received:	16-Apr-2025
Accepted:	23-Apr-2025

Senior Editor: Paul Greenhaff

Reviewing Editor: Paul Greenhaff

Transaction Report:

Dear Dr Racinais,

Re: JP-RP-2024-287777 "Muscle regeneration is improved by hot water immersion but unchanged by cold following a simulated musculoskeletal injury in humans" by Valentin Dablainville, Adele Mornas, Tom Normand-Gravier, Maha Al-Mulla, Emmanouil Papakostas, Bruno Olory, Theodorakys Marín Fermín, Frantzeska Zampeli, Nelda Nader, Marine Alhammoud, Freya Bayne, Anthony Sanchez, Marco Cardinale, Robin Candau, Henri Bernardi, and Sebastien Racinais

Thank you for submitting your manuscript to The Journal of Physiology. It has been assessed by a Reviewing Editor and by 2 expert referees and we are pleased to tell you that it is potentially acceptable for publication following satisfactory major revision.

REVISION CHECKLIST:

We look forward to receiving your revised submission.

Yours sincerely,

Paul Greenhaff
Senior Editor
The Journal of Physiology

EDITOR COMMENTS

Reviewing Editor:

Comments to the Author:

This work is novel and interesting and will advance the current understanding of the field. Both reviewers agree that the article holds promise but note some addressable concerns in the writing and discussion of the results. Addressing these as well other reviewer comments will strengthen the manuscript and improve its suitability for publication.

Senior Editor:

Comments to the Author:

Thank you for the manuscript submission to The Journal of Physiology. It has been considered by two expert reviewers and a reviewing editor. All concerned can see novelty in the workout but are also of the opinion that the manuscript largely needs to be rewritten. This can be addressed, but the authors should not take the concern of the reviewers lightly on this matter. More problematic are the major issues perceived by Referee 2 about the study design and robustness of the experimental data, which currently detracts in a major way from the veracity of the conclusions made by the authors. This needs very careful consideration by the authors and inclusion of new analysis and data to address some of the concerns raised is suggested. This will improve the manuscript and thereby increase the priority for publication in The Journal of Physiology.

REFEREE COMMENTS

Referee #1:

Thanks for the opportunity to offer feedback on "Muscle regeneration is improved by hot water immersion but unchanged by cold following a simulated musculoskeletal injury in humans." This is an interesting paper. The authors should be praised for their considerable time and effort in executing this study design. The need for the current study is well articulated and the data certainly add to current understanding.

Overall, the writing needs further polishing, and I have tried to help identify issues throughout, particularly relating to punctuation and singular/plural word forms. A redrafting of the Discussion is suggested to improve its structure and cohesion.

It is hoped that the following comments will be useful.

Specific feedback

Line 80: The Introduction demonstrates a sound understanding of the current literature, as the justification for this study is

well presented.

Line 84: "the involved processes of muscle regeneration" reads awkwardly. Changing to 'the processes involved in muscle regeneration' or similar would alleviate this issue.

Line 87: Should "mechanism" be plural?

Line 87: A comma is missing before "such as" and after "angiogenesis". Please add.

Line 88: Should "tissues" be singular?

Line 95: A comma is missing before "often". Please add.

Line 99: A comma is missing before and after "to our knowledge". Please add.

Line 114: Should "limit" be plural?

Line 133: "in human as compared to a control group" may read more clearly if worded 'in humans compared to a control group'.

Line 138: Please add 'and' before "2)" to improve fluency.

Line 144: Should the specific hospital be named for transparency?

Line 150: Please justify this sample size.

Line 150: Notably, the participants are all male. Sex does not appear to be an exclusion criterion. Can the authors please clarify this situation?

Line 154: Current fitness and/or training status is relevant. Could these descriptors be added to assist reader understanding of the cohort?

Line 155: "time period" is redundant. For conciseness, please use 'time' or 'period' only.

Line 157: Please define "abnormal values" of CK and Mb to avoid reader misinterpretation.

Line 163: Please correct "at familiarization phase" to 'at the familiarization phase'.

Line 164: Did the nutritional interventions actually standardize energy and protein intake or help to achieve this outcome? The latter is interpreted as a more accurate reflection. If correct, please edit the wording accordingly.

Lines 166-168: Was compliance with the protein, supplements, and alcohol requirements assessed?

Line 170: Were exclusion criteria placed on any medications? Further, were participants remunerated for their involvement?

Line 173: "figure" should be capitalized in this context. Please correct.

Line 176: While US spelling is mostly used, there are also instances of British spelling (e.g., familiarisation) throughout. Please review for consistency and alignment with the Journal's requirements.

Line 174: A comma is missing before "which". Please add.

Line 176: A comma is missing before "including". Please add.

Line 185: Were lab testing times matched throughout the 11 days?

Line 194: There is either a punctuation, conjunction, or word choice issue relating to "settled". Please address.

Line 216: Immersion to the waist effectively targets a change in muscle temperature versus a change in core body temperature. Was this the intention, or were other factors involved?

Lines 218-220: Exposure durations to the hydrostatic pressure of the water immersion differ considerably between conditions. Is this a limitation and/or concern?

Lines 240-241: Please define P.E and P.I.

Line 244: Pain and soreness are different constructs. Accordingly, is it appropriate to measure muscle soreness with a muscle pain visual analogue scale?

Line 242-247: Please detail the validity and reliability of the visual analogue scale in the used context.

Line 250-251: Curiously, the Biodex induced the simulated muscle injury, but a different device was used to measure strength. Can this reason please be explained?

Lines 25-260: A comma is missing before and after "each lasting 3 to 5 seconds". Please add.

Line 269: What was the period between collection and analysis, and how were the samples treated in the meantime (e.g., centrifugation, refrigeration/freezing, etc.)?

Line 273: Please present the intra-assay (and inter-assay if appropriate) reliability data for this analyses.

Line 276: Please consider changing "one time" to 'once' for conciseness.

Line 276: These thermal measures are invasive and/or expensive. Accordingly, it is understandable that their measurement frequency would be minimised. However, even as a descriptive measure, is there any risk of adaptation over the intervention period that might skew the data? Might this be somewhat overcome by matching the participant numbers and measurement timings between groups?

Line 279: A further explanation for why the unexercised leg was measured would be insightful. In particular, the authors are confident that this muscle temperature matches that in the exercised leg, given the change in inflammation, metabolism, and blood flow that the simulated muscle injury would cause.

Line 280: A recent paper highlighted the effect of varying subcutaneous fat and muscle thickness on muscle temperature measures (<https://www.sciencedirect.com/science/article/pii/S0306456524001438>). Can the authors please outline their approach to measuring and standardizing muscle temperature measurement depth?

Lines 283-284: Please add details about the calibration and ingestion of the core body temperature pills.

Lines 298-299: How long were samples stored before analysis?

Line 301: Were meals planned "with" or 'by' the dietician?

Line 313: Please add 'and' before "transferred".

Line 337: "analysed" is another example of British English.

Line 356: Was sphericity also assessed?

Line 368: Please consider including effect sizes and 95% Confidence Intervals alongside the linear mixed models. This would enhance the research findings' interpretability, contextual relevance, transparency, and practical significance.

Line 439: Consideration is required regarding the current Discussion structure and its effectiveness in clearly and concisely exploring and explaining the findings. This is a large study (and a long Discussion), and it is expected that this is why the authors have chosen to use sub-headings throughout this section. However, instead of discussing variables in isolation, consider integrating the findings to provide a more holistic view of how cold and hot treatments affect muscle regeneration. This can help highlight the interconnectedness of the different physiological responses and address the flow throughout the Discussion, which is not always smooth. Subheadings are ok, but integrating variables to present themes would be helpful, particularly regarding the molecular and cellular responses. Accordingly, instead of separate sections on individual markers, integrating the findings and presenting on broader themes will make the Discussion more readable. It is understood that this represents considerable redrafting, but it is also expected that the Discussion would be clearer, more concise, and make a greater contribution to the current literature.

Line 455: Please change "add" to the plural form.

Line 463: Please add a comma before "potentially".

Line 470: Please add a comma before "potentially".

Line 472: Please correct "using CWI or HWI as recovery intervention" to 'using CWI or HWI as a recovery intervention'.

Line 476: A comma is missing before "suggesting". Please add.

Line 479: "e.g" is incorrectly punctuated. Please edit.

Line 481: "treatment" should be plural. Please change.

Line 489: A comma is missing before "with". Please add.

Line 498: The comma after "concentration" is unnecessary. Please delete.

Line 499: A comma is missing before "promoting". Please add.

Line 511: Please change "human" to the plural form.

Line 513: The comma after "cells" is unnecessary. Please delete.

Line 513: "play a role" or 'play the role'?

Line 516: Should "by" be 'in'?

Line 520: Please change "human" to the plural form.

Line 524: "minutes" needs to be changed to the singular form. Please correct.

Line 533: A comma is missing before "particularly". Please add.

Line 536: A comma is missing before "although". Please add.

Line 559: "et al" is to be incorrectly punctuated. Please correct.

Line 559: The comma after "(2021)" is unnecessary. Please remove.

Line 562: The plural verb "have" does not agree with the singular subject "no specific study". Please correct the subject-verb alignment.

Line 578: A comma is missing before "although". Please add.

Line 599: A comma is missing after "However". Please add.

Line 600: A comma is missing before and after "while COLD ". Please add.

Line 600: A comma is missing after "efficiency". Please add.

Line 610: 'with' might be a better word than "to" in this sentence.

Line 623: "longer period" is missing a determiner before it. Please add an article.

Line 625: "placebo effect" is missing a determiner before it. Please add an article.

Lines 628-629: Should "the benefits of heat exposures unlikely due to a placebo effect" be 'the benefits of heat exposures are unlikely due to a placebo effect'?

Line 630: "sham modality" is missing a determiner before it. Please add an article.

Lines 635-638: This sentence is long and lacking punctuation. Please review.

Line 643: "et al" is incorrectly punctuated. Please address.

Line 647: "et al" is incorrectly punctuated. Please address.

Referee #2:

GENERAL COMMENTS

I commend the authors on their work. These types of studies are time-consuming and logistically challenging. The authors have addressed an important issue in sports medicine and they have attempted to address some of the controversies in the existing literature.

Impact: This study addresses an important problem about strategies to promote regeneration of muscle following extensive muscle injury.

Insights into physiological mechanisms: The authors have combined subjective, perceptual variables with objective quantitative and mechanistic variables. In this regard, the study is reasonably comprehensive.

Originality: The study is original for several reasons. As stated above, it includes a wider range of variables compared with most other research in this field. It compares two common treatments (i.e., cold water immersion and hot water immersion) under similar conditions of muscle damage. Most other research has only examined these treatments in isolation. The time course of investigation is longer than most other research.

Study design and robustness of the experimental data: Unfortunately, the study design is weakened because the duration of the three treatments was very different. Cold water immersion was 15 minutes, thermoneutral water immersion was 30 minutes and hot water immersion was 60 minutes. The authors indicate that they designed these protocols based on the literature. This is understandable to some degree. However, the differences in duration of treatment could have influenced the study outcomes independently of water temperature. A better approach may have been to compare the effects of 15 minutes of each treatment (as cold water immersion is arguably the least tolerable treatment). The muscle damage protocol induced severe muscle damage, as indicated by the sustained decrement in MVC strength. However, it is uncertain how well this protocol applies to 'real-world' scenarios that athletes may face. Because muscle biopsies are invasive, it is often difficult to obtain more than three or four biopsies from individuals. That said, the rationale for the timing of muscle biopsies in the present study. The authors may have missed the peak in some variables in the muscle tissue. The authors presented some histology and immunohistochemistry images. However, they did not conduct any quantitative analysis of the number/cross-sectional areas of regenerating muscle fibres. Nor did they compare the images between the treatment groups. Because muscle regeneration was the key aspect of this study, this lack of analysis is an obvious omission. It would have been useful to measure the number of leucocytes and satellite cells in the muscle biopsies as an indication of inflammation and muscle regeneration, respectively. The authors should discuss the issues as limitations of their study.

Validity of the conclusions: The authors conclude overall that compared with cold water, hot water immersion may be more favourable for promoting muscle regeneration. This conclusion was based on findings that hot water immersion reduced muscle soreness and circulating muscle damage markers. Unfortunately, there is not really enough data to support the conclusion that hot water immersion reduced pro-inflammatory responses and enhanced anti-inflammatory responses. This conclusion was based only on measurements of NF- κ B and IL-10, and only two biopsies at day 5 and 11.

SPECIFIC COMMENTS:

The manuscript has quite a few grammatical and formatting errors. It would benefit from editing by a native English speaker.

The use of different terms such as cold water immersion, COLD, cryotherapy, HWI and HOT disrupts the readability of the manuscript. It would be better to use one or two terms consistently throughout the manuscript.

In lines 157-158, the authors stated that they excluded anyone with abnormally high CK activity or myoglobin concentration. It would be useful if the authors indicated how many people were excluded on this basis.

As there were 34 people who completed the study, and three groups, the number of people per group should be updated (i.e., from n=12/group) throughout the manuscript.

Line 298: RNA extraction is mentioned, but there is no mRNA data presented in the manuscript.

Figure 2 requires a scale bar on these images.

Line 375: This subheading doesn't see appropriate for all of the data presented in this paragraph.

Lines 443-444. The association between muscle soreness and VEGF expression is unclear.

Lines 462-464. This conclusion seems quite superficial.

Lines 472-474. This statement is not very accurate, as there are a number of studies that have reported improvements in muscle strength following cold water immersion.

Lines 480-482. This conclusion is very vague.

Lines 497-500. This conclusion seems quite superficial, and it doesn't attempt to tie in the data around HSPs.

Lines 579-584. There are some conflicting statements here that require more careful thought. How might reduced VEGF expression assist muscle regeneration?

Lines 635-638. There are some conflicting statements here that require more careful thought.

Lines 641-643. This sentence is confusing.

The references are not formatted consistently.

Because TGF-beta1 did not change, it is debatable whether it should be mentioned in the discussion.

The manuscript is missing some key papers that should be included for a more balanced discussion (e.g., Roberts et al. 2015 J Physiol 593: 4285-4301; Peake et al 2017 J Physiol 595: 695-711; D'Souza et al 2018 Am J Physiol 314: R824-R833; Peake et al 2020 Front Physiol 11: 737.

END OF COMMENTS

EDITOR COMMENTS

Reviewing Editor:

Comments to the Author:

This work is novel and interesting and will advance the current understanding of the field. Both reviewers agree that the article holds promise but note some addressable concerns in the writing and discussion of the results. Addressing these as well other reviewer comments will strengthen the manuscript and improve its suitability for publication.

AR: The authors thank the reviewing editor for the positive feedback. We have integrated all reviewer suggestions and hope that the manuscript is now clearer. The modifications are indicated in the revised text in red font and our replies are included below.

Senior Editor:

Comments to the Author:

Thank you for the manuscript submission to The Journal of Physiology. It has been considered by two expert reviewers and a reviewing editor. All concerned can see novelty in the workout but are also of the opinion that the manuscript largely needs to be rewritten. This can be addressed, but the authors should not take the concern of the reviewers lightly on this matter. More problematic are the major issues perceived by Referee 2 about the study design and robustness of the experimental data, which currently detracts in a major way from the veracity of the conclusions made by the authors. This needs very careful consideration by the authors and inclusion of new analysis and data to address some of the concerns raised is suggested. This will improve the manuscript and thereby increase the priority for publication in The Journal of Physiology.

AR: The authors thank the senior editor for the opportunity to amend this manuscript. We have integrated all reviewer suggestions and hope that the manuscript is now clearer. As suggested, the discussion has also been largely rewritten.

REFEREE COMMENTS

Referee #1:

Thanks for the opportunity to offer feedback on "Muscle regeneration is improved by hot water immersion but unchanged by cold following a simulated musculoskeletal injury in humans." This is an interesting paper. The authors should be praised for their considerable time and effort in executing this study design. The need for the current study is well articulated and the data certainly add to current understanding.

Overall, the writing needs further polishing, and I have tried to help identify issues throughout, particularly relating to punctuation and singular/plural word forms. A redrafting of the Discussion is

suggested to improve its structure and cohesion.

It is hoped that the following comments will be useful.

AR: The authors thank the reviewer for the useful suggestions. The modifications are indicated in the revised text in red font and our replies are included bellow. We also checked carefully the punctuation and singular/plural forms. Moreover, as suggested, the discussion has also been redrafted.

Specific feedback

Line 80: The Introduction demonstrates a sound understanding of the current literature, as the justification for this study is well presented.

AR: The authors thank the reviewer for this positive comment.

Line 84: "the involved processes of muscle regeneration" reads awkwardly. Changing to 'the processes involved in muscle regeneration' or similar would alleviate this issue.

AR: Modified to 'the processes involved in muscle regeneration'

Line 87: Should "mechanism" be plural?

AR: This has been corrected to read "Mechanisms"

Line 87: A comma is missing before "such as" and after "angiogenesis". Please add.

AR: Added

Line 88: Should "tissues" be singular?

AR: Modified by "tissue"

Line 95: A comma is missing before "often". Please add.

AR: Added

Line 99: A comma is missing before and after "to our knowledge". Please add.

AR: Added

Line 114: Should "limit" be plural?

AR: Modified by "limits"

Line 133: "in human as compared to a control group" may read more clearly if worded 'in humans compared to a control group'.

AR: Modified to "in humans compared to a control group"

Line 138: Please add 'and' before "2)" to improve fluency.

AR: Added

Line 144: Should the specific hospital be named for transparency?

AR: The name of the hospital has been added and the sentence modified: "Aspetar Orthopaedic and Sport Medicine Hospital scientific committee (#ASC/0000245/ak) and by Aspire Zone Foundation institutional review board"

Line 150: Please justify this sample size.

AR: As no human study previously investigated the effect of muscle cooling or heating during a muscle injury, pilot testing performed by our team were used to compute the necessary sample size. G*Power statistical analyses software (version 3.1.9.7, Germany) was used to calculate the necessary sample size for 3 groups based on a force output measured 4 times, using a 0.05 α -error, and a 0.9 β -error. The total sample size calculated by the software was 27 participants (9/group) and it was decided to increase the sample size to 12/group to compensate eventual drops of participants.

Added in the text: "As computed using G*Power statistical analyses software (version 3.1.9.7, Germany) for an output measured 4 times in 3 groups, a sample size of 27 participants (9/group) was required with a 0.05 α -error, and a 0.9 β -error. The sample size was increased to 12/group to compensate eventual drops of participants."

Line 150: Notably, the participants are all male. Sex does not appear to be an exclusion criterion. Can the authors please clarify this situation?

AR: The recruitment was open to male and female participants. However, during the initial participants recruitment, only one female volunteered to participate to this experiment making impossible a proper randomisation between groups. Furthermore, large differences between males and females have recently been observed on several aspects of post-injury muscle regeneration in humans and animals (Luk et al., 2021, 10.3389/fphys.2021.752347, You et al., 2023, 10.14814/phy2.15791). Therefore, considering the differences in muscle regeneration between sexes and the very low number of potential female participants, it was decided to include only male participants to reduce results variability.

The following was added to the manuscript: "Of note, this research project was open to males and females participants but only one female volunteered to participate. Therefore, considering the large differences in muscle regeneration recently reported between sexes and the limited number of potential female participants for equal repartition between groups, it was decided to include male participants only during this study (Luk et al., 2021; You et al., 2023)."

Line 154: Current fitness and/or training status is relevant. Could these descriptors be added to assist reader understanding of the cohort?

AR: The following was added to the manuscript: "and maintaining a recreative level of training"

Line 155: "time period" is redundant. For conciseness, please use 'time' or 'period' only.

AR: Modified by "during the time of the study"

Line 157: Please define "abnormal values" of CK and Mb to avoid reader misinterpretation.

AR: The range of normal values has been added and the sentence has been modified for more clarity: "Circulating Creatine-Kinase and Myoglobin levels were assessed during the week preceding the start of the experiment to ensure that all participants had values within the normal ranges (31-936U/L for CK and 0-72 μ g/L for myoglobin)."

Line 163: Please correct "at familiarization phase" to 'at the familiarization phase'.

AR: Modified by "at the familiarization phase"

Line 164: Did the nutritional interventions actually standardize energy and protein intake or help to achieve this outcome? The latter is interpreted as a more accurate reflection. If correct, please edit the wording accordingly.

AR: Modified in the manuscript: "The nutritional interventions promoted a standardized energy and protein intake without setting absolute targets"

Lines 166-168: Was compliance with the protein, supplements, and alcohol requirements assessed?

AR: A strict compliance with the requirements was not possible to assess during this study. The participants had two nutritional consultations with a sport dietician to define their nutritional requirements and standardize the energy and protein intake. At this occasion they were reminded on the further requirements of the study (stop nutritional complements/supplements and alcohol) and general nutritional instructions were repeated across the experiments by the researchers.

Modified in the manuscript: "Participants were instructed, both during the nutritional interventions and on several other occasions by the research team, not to consume any supplementations (e.g. whey proteins, creatine), as well as alcohol and pain medication during the study period."

Line 170: Were exclusion criteria placed on any medications? Further, were participants remunerated for their involvement?

AR: It has been clarified in the revised manuscript that "Exclusion criteria included prescribed medication for chronic medical condition. Furthermore, participants were instructed to refrain from taking any pain killers or anti-inflammatories." Moreover, it has also been indicated that "Due to the burden associated with simulated muscle injury and repeated muscle biopsies, participants were financially compensated for their participation in this study."

Line 173: "figure" should be capitalized in this context. Please correct.

AR: Modified by "FIGURE"

Line 176: While US spelling is mostly used, there are also instances of British spelling (e.g., familiarisation) throughout. Please review for consistency and alignment with the Journal's requirements.

AR: UK English is now used throughout the manuscript.

Line 174: A comma is missing before "which". Please add.

AR: Added

Line 176: A comma is missing before "including". Please add.

AR: Added

Line 185: Were lab testing times matched throughout the 11 days?

AR: All the participants reported to the lab at different times of the day according to their personal schedule and to the limited availability of the testing material. However, each participant reported daily at the same time during the study to complete the tests and the assigned thermal intervention. Only the biopsies were sampled for all the participants between 6 and 8am.

Added in the text: "Participants reported to the lab daily at the same time of the day, during the 11 days following the muscle damage protocol to complete testing, their assigned thermal intervention, blood sample collections and muscle biopsies. Of note, only the biopsies were sampled for all participants between 6 and 8am, with the participants fasting."

Line 194: There is either a punctuation, conjunction, or word choice issue relating to "settled". Please address.

AR: Modified by "set"

Line 216: Immersion to the waist effectively targets a change in muscle temperature versus a change in core body temperature. Was this the intention, or were other factors involved?

AR: This study was intended to target only a change in muscle temperature. As this investigation focused on the effect of local heating or cooling on muscle recovery/regeneration, minimal changes in core temperature were intended in order to reduce potential systemic effect with core temperature elevations.

Added in the manuscript: "The thermal interventions used in this study were designed to modify muscle temperature and local blood flow, with minimal changes in core body temperature. "

Lines 218-220: Exposure durations to the hydrostatic pressure of the water immersion differ considerably between conditions. Is this a limitation and/or concern?

AR: Thank you to point out this aspect of the immersion that we did not discuss in the manuscript.

To our knowledge, no conclusive evidence exists regarding the effect of the hydrostatic pressure on muscle recovery/regeneration (Versey et al., 2013). Moreover, even if this effect was unlikely, we further reduced it by immersing the control group also. Thus, while it is not possible to ascertain that hydrostatic pressure did not have any effect on muscle recovery from the simulated injury, the authors are confident that the selected procedure minimized potential bias.

Moreover, this point has been added to the discussion as a limitation: "However, it should be noted that the differences in immersion durations did not allow to estimate whether the effects were specific to the involved temperature or in combination with hydrostatic pressure. Despite very unlikely, it is not possible to ascertain that hydrostatic pressure did not have any effect on muscle recovery from the simulated injury (Wilcock et al., 2006; Versey et al., 2013)."

Lines 240-241: Please define P.E and P.I.

AR: Modified in the text: "immediately post muscle damage (Post-exercise at D0), after the first bath (Post-immersion at D0)"

Line 244: Pain and soreness are different constructs. Accordingly, is it appropriate to measure muscle soreness with a muscle pain visual analogue scale?

AR: Thank you to point out this confusion.

The terminology has been replaced by “pain” throughout the manuscript and the figures.

Line 242-247: Please detail the validity and reliability of the visual analogue scale in the used context.

AR: Pain was measured using a visual analogue scale representing the pain perception in a straight line from no pain to worst pain perception (0 – 20). No graduation was visible by the participant. The VAS is considered as a reference measure in the evaluation of pain perception and is used since 1921, and is still commonly used particularly in clinical settings (Delgado *et al.*, 2018). This is a quick and well accepted measure of pain by the participant. A high reliability of the VAS measure of pain has been shown in acute abdominal pain (Gallagher *et al.*, 2002) and a moderate to good reliability in patients with chronic musculoskeletal pain (Boonstra *et al.*, 2008). In the absence of objective gold standard measurement for pain, the validity of the VAS cannot be evaluated. However, in osteoarthritic patients, significant correlations between VAS scores and demographic variables such as age, BMI, and osteoarthritis grade have been found supporting the validity of the VAS in pain measurement (Alghadir *et al.*, 2018). VAS have previously been used in studies using similar muscle damage protocol (Cramer *et al.*, 2004, 2007; Mackey *et al.*, 2016).

Added in the manuscript: “A visual analogue scale (VAS) was used to assess pain perception, similarly to previous studies using electrically stimulated eccentric muscle damage (Cramer *et al.*, 2004, 2007; Mackey *et al.*, 2016). The VAS used in this study was graduated from 0 (no pain) to 20 (worst pain). No graduation was visible by the participant during the measurement.”

Line 250-251: Curiously, the Biodex induced the simulated muscle injury, but a different device was used to measure strength. Can this reason please be explained?

AR: The Biodex 3 machine was only used in eccentric mode to induce the simulated muscle injury. A dedicated isometric knee extension ergometer presenting a higher rigidity than the Biodex machine was used to measure strength. Furthermore, the isometric ergometer presented a higher sampling rate (2000Hz), increasing measurement accuracy.

Modified in the manuscript:” The isometric knee extensor force of the right leg was measured using a high-rigidity custom-made dynamometric chair.”

Lines 25-260: A comma is missing before and after "each lasting 3 to 5 seconds". Please add.

AR: Added

Line 269: What was the period between collection and analysis, and how were the samples treated in the meantime (e.g., centrifugation, refrigeration/freezing, etc.)?

AR: All samples were collected on site by a phlebotomist and centrifuged for 10min at 3000RPM.

CK was processed by the hospital laboratory in the 2 to 3 hours following sampling. The Dimension Vista 500 system (Siemens, Germany) was used to quantify CK.

Myoglobin was referred to an external laboratory (Bioscientia, Germany). Samples were separated from red blood cells, stored at a temperature of 2-8°C and shipped in a cool box. The turnaround time (from collection to results) for this test was 3 days.

Added in the text:

“All samples were centrifuged 10min at 3000RPM.” And “in the 2 to 3 hours following sampling”.

“For myoglobin (Mb), plasma was separated from red blood cells, stored at a temperature of 2-8°C and shipped in a cool box to an accredited partner laboratory (Bioscientia, Germany) where specimens were

core temperature to only 38°C, lower than the core temperature previously reported to induce heat acclimation (Périard *et al.*, 2015; Pryor *et al.*, 2019).

Added in the manuscript: “In contrast, while the single core temperature measurement performed during this study does not allow definitive conclusions, the available literature suggests that the thermal interventions were unlikely to have triggered a cold or heat acclimation response during the experiment (Périard *et al.*, 2015; Gordon *et al.*, 2019).”

Line 279: A further explanation for why the unexercised leg was measured would be insightful. In particular, the authors are confident that this muscle temperature matches that in the exercised leg, given the change in inflammation, metabolism, and blood flow that the simulated muscle injury would cause.

AR: We thank the reviewer to point out this specific aspect that needs further clarifications.

The muscle temperature is an invasive act that requires to insert a 3cm needle in the muscle belly during approximately 10-15 seconds. In this study, it was decided to assess only the unexercised leg for 2 reasons:

- Some of the participants experienced high muscular pain until the end of the experiment. Despite a superficial anaesthesia (mostly skin and underlying superficial fat tissue) applied during the procedure, the muscle remained painful when the procedure was performed on the injured leg. It was then decided to limit the measurement to the uninjured leg to decrease the pain experienced by the participants and facilitate the measurement.

- Recent evidence examining the effect of repeated biopsies showed that a previous biopsy can interfere with the outcome found in a subsequent biopsy (Long *et al.*, 2023). The potential modifications in the regeneration processes created by the muscle temperature measurement remain unknown, however, to limit potential interferences with biopsy results and avoid disrupting regenerative processes, it was decided to perform the muscle temperature measurement in the unexercised leg only.

A point of discussion was added in the limits section at the end of the manuscript: “It should be acknowledged that the muscle temperature measurement was performed only on the unexercised limb to reduce participant discomfort in the injured leg and avoid potential interference between subsequent biopsies from the injured leg (Long *et al.*, 2023). However, the effect of the simulated injury on the intramuscular temperature at the end of the intervention remains unknown. Indeed, potential differences in muscle temperature between the exercised and unexercised limbs, which were not accounted for during this study, could exist.”

Line 280: A recent paper highlighted the effect of varying subcutaneous fat and muscle thickness on muscle temperature measures (<https://www.sciencedirect.com/science/article/pii/S0306456524001438>). Can the authors please outline their approach to measuring and standardizing muscle temperature measurement depth?

AR: We thank the reviewer for his suggestions on muscle temperature.

In our study, muscle temperature was measured once using a thermo-sensor needle marked for measurement at 3, 2 and 1cm depths. Once the measurement site prepped (local anesthesia and sanitizing), the thermos-sensor needle was inserted successively at 3, 2 and 1cm under the skin and kept at each depth for approximately 4 seconds. Subcutaneous fat tissue and muscle thickness were not accounted for during this measurement.

Added in the text: “Muscle temperature was assessed only in the unexercised leg to avoid interfering with the regenerative processes of the exercised leg and to limit pain levels experienced by the participants, as the exercised muscle remained painful until the end of the experiment. Measurements were recorded at 3, 2 and 1cm muscle depth under the skin with the thermo-sensor needle left at each depth for approximately 4 seconds to ensure stable temperature value. Of note, unlike Rodrigues et al. (2024), ultrasound measurements of subcutaneous fat tissue and muscle thickness were not performed (Rodrigues et al., 2024).”

Lines 283-284: Please add details about the calibration and ingestion of the core body temperature pills.

AR: The details communicated by Bodycap company have been added to the text: “The core temperature pill was ingested with water at least 4 hours before the start of the hot water immersion and 1 hour before the start of cold or neutral immersion. While the HOT group was allowed to drink *ad libitum* during the water immersion, the COLD and NEUTRAL groups were instructed to refrain from eating or drinking until core temperature measurement was achieved. The e-celsius pills contain a factory-calibrated sensor, ensuring $\pm 0.1^{\circ}\text{C}$ precision, and are verified in a temperature-controlled bath to ensure their conformity to the targeted temperature range of 36-41°C.”

Lines 298-299: How long were samples stored before analysis?

AR: The samples were stored for 6 months to 1 year before analysis.

Modified in manuscript by: “All samples were stored at -80°C for 6 months to 1 year being analysed.”

Line 301: Were meals planned "with" or 'by' the dietician?

AR: Modified in the text by “by”

Line 313: Please add 'and' before "transferred".

AR: Added

Line 337: "analysed" is another example of British English.

AR: Changed to British spelling style

Line 356: Was sphericity also assessed?

AR: Sphericity of the results was not assessed in this study as it is not a required assumption to perform linear mixed models or non-parametric statistical tests.

Magezi et al, 2015, doi: 10.3389/fpsyg.2015.00002

Line 368: Please consider including effect sizes and 95% Confidence Intervals alongside the linear mixed models. This would enhance the research findings' interpretability, contextual relevance, transparency, and practical significance.

AR: Thank you to point out this omission from the Statistical methodology.

Added in the manuscript: “The difference of the means, 95% confidence intervals of the differences between groups or between times, and Hedges G effect sizes (ES) are included alongside the p-values calculated using LMM. The interpretation of the Hedges G effect sizes is as follows: small effect=0.2, medium effect=0.5, large effect=0.8 (Brydges, 2019).”

Line 439: Consideration is required regarding the current Discussion structure and its effectiveness in clearly and concisely exploring and explaining the findings. This is a large study (and a long Discussion), and it is expected that this is why the authors have chosen to use sub-headings throughout this section. However, instead of discussing variables in isolation, consider integrating the findings to provide a more holistic view of how cold and hot treatments affect muscle regeneration. This can help highlight the interconnectedness of the different physiological responses and address the flow throughout the Discussion, which is not always smooth. Subheadings are ok, but integrating variables to present themes would be helpful, particularly regarding the molecular and cellular responses. Accordingly, instead of separate sections on individual markers, integrating the findings and presenting on broader themes will make the Discussion more readable. It is understood that this represents considerable redrafting, but it is also expected that the Discussion would be clearer, more concise, and make a greater contribution to the current literature.

AR: We thank the reviewer for this suggestion that we hope, considerably improved the quality of the manuscript. The discussion has been largely redrafted to improve the integration of the findings in the perspective of the literature of cold and hot.

Line 455: Please change "add" to the plural form.

AR: Modified: "adds"

Line 463: Please add a comma before "potentially".

AR: Added

Line 470: Please add a comma before "potentially".

AR: Added

Line 472: Please correct "using CWI or HWI as recovery intervention" to 'using CWI or HWI as a recovery intervention'.

AR: Corrected, the "a" has been added.

Line 476: A comma is missing before "suggesting". Please add.

AR: Added

Line 479: "e.g" is incorrectly punctuated. Please edit.

AR: Modified to "e.g.,"

Line 481: "treatment" should be plural. Please change.

AR: Changed to plural.

Line 489: A comma is missing before "with". Please add.

AR: Added

Line 498: The comma after "concentration" is unnecessary. Please delete.

AR: Removed

Line 499: A comma is missing before "promoting". Please add.

AR: Added

Line 511: Please change "human" to the plural form.

AR: Modified to plural for the entire manuscript.

Line 513: The comma after "cells" is unnecessary. Please delete.

AR: Removed

Line 513: "play a role" or 'play the role'?

AR: Modified by "play the role".

Line 516: Should "by" be 'in'?

AR: Modified by "in".

Line 520: Please change "human" to the plural form.

AR: Modified by "humans".

Line 524: "minutes" needs to be changed to the singular form. Please correct.

AR: Modified by "minute".

Line 533: A comma is missing before "particularly". Please add.

AR: Added

Line 536: A comma is missing before "although". Please add.

AR: Added

Line 559: "et al" is to be incorrectly punctuated. Please correct.

AR: Modified to "Kawashima *et al.* (2021)"

Line 559: The comma after "(2021)" is unnecessary. Please remove.

AR: Modified to "Kawashima *et al.* (2021)" and coma has been removed.

Line 562: The plural verb "have" does not agree with the singular subject "no specific study". Please correct the subject-verb alignment.

AR: Modified to singular "has".

Line 578: A comma is missing before "although". Please add.

AR: Added

Line 599: A comma is missing after "However". Please add.

AR: Added

Line 600: A comma is missing before and after "while COLD ". Please add.

AR: Added

Line 600: A comma is missing after "efficiency". Please add.

AR: Added

Line 610: 'with' might be a better word than "to" in this sentence.

AR: Modified to "with".

Line 623: "longer period" is missing a determiner before it. Please add an article.

AR: "a" was added to the sentence.

Line 625: "placebo effect" is missing a determiner before it. Please add an article.

AR: "a" was added to the sentence.

Lines 628-629: Should "the benefits of heat exposures unlikely due to a placebo effect" be 'the benefits of heat exposures are unlikely due to a placebo effect'?

AR: Modified to "are unlikely due".

Line 630: "sham modality" is missing a determiner before it. Please add an article.

AR: "a" was added to the sentence.

Lines 635-638: This sentence is long and lacking punctuation. Please review.

AR: This sentence has been removed.

Line 643: "et al" is incorrectly punctuated. Please address.

AR: Modified to "et al."

Line 647: "et al" is incorrectly punctuated. Please address.

AR: Modified to "et al."

Referee #2:

GENERAL COMMENTS

I commend the authors on their work. These types of studies are time-consuming and logistically challenging. The authors have addressed an important issue in sports medicine, and they have attempted to address some of the controversies in the existing literature.

Impact: This study addresses an important problem about strategies to promote regeneration of muscle following extensive muscle injury.

Insights into physiological mechanisms: The authors have combined subjective, perceptual variables with objective quantitative and mechanistic variables. In this regard, the study is reasonably

comprehensive.

Originality: The study is original for several reasons. As stated above, it includes a wider range of variables compared with most other research in this field. It compares two common treatments (i.e., cold water immersion and hot water immersion) under similar conditions of muscle damage. Most other research has only examined these treatments in isolation. The time course of investigation is longer than most other research.

AR: The authors thank the reviewer for the positive feedback. We have integrated all suggestions and hope that the manuscript is now clearer. The modifications are indicated in the revised text in red font and our replies are included bellow.

Study design and robustness of the experimental data: Unfortunately, the study design is weakened because the duration of the three treatments was very different. Cold water immersion was 15 minutes, thermoneutral water immersion was 30 minutes and hot water immersion was 60 minutes. The authors indicate that they designed these protocols based on the literature. This is understandable to some degree. However, the differences in duration of treatment could have influenced the study outcomes independently of water temperature. A better approach may have been to compare the effects of 15 minutes of each treatment (as cold-water immersion is arguably the least tolerable treatment).

AR: Thank you to point out these aspects of the protocol that appear unclear.

To our knowledge, no conclusive evidence exists regarding the effect of the hydrostatic pressure on muscle recovery/regeneration (Wilcock *et al.*, 2006; Versey *et al.*, 2013). Moreover, even if this effect was unlikely, we further reduced it by immersing the control group also. Thus, while it is not possible to ascertain that hydrostatic pressure did not have any effect on muscle recovery from the simulated injury, the authors are confident that the selected procedure minimized potential bias.

Moreover, this point has been added to the discussion as a limitation: "However, it should be noted that the differences in immersion durations did not allow to estimate whether the effects were specific to the involved temperature or in combination with hydrostatic pressure. Despite very unlikely, it is not possible to ascertain that hydrostatic pressure did not have any effect on muscle recovery from the simulated injury (Versey *et al.*, 2013)."

Furthermore, a similar immersion duration (15min) in all groups, as suggested, would have elicited a different muscle temperature, especially in HWI. Based on the literature on heat, achieving a sufficient muscle temperature stimulus could be key to induce or not some effects in the muscle (Nosaka *et al.*, 2006; Kakigi *et al.*, 2011; Kim *et al.*, 2019a; Hafen *et al.*, 2019; Fuchs *et al.*, 2020; Ihsan *et al.*, 2020; Sautillet *et al.*, 2023, 2024).

The muscle damage protocol induced severe muscle damage, as indicated by the sustained decrement in MVC strength. However, it is uncertain how well this protocol applies to 'real-world' scenarios that athletes may face.

AR: The different forms of "real-world" traumatic human skeletal muscle damage do not allow for the study of muscle regeneration in a systematically standardised way (Mackey & Kjaer, 2017). Previous studies on cryotherapy and heat, mainly investigated their effect on the recovery of voluntary lengthening contractions that rarely lead to myofiber necrosis. Despite this model being valid to assess muscle recovery, it appears to elicit different processes than a traumatic muscle injury. In contrast, the model used in this study has been shown to induce significant myofiber necrosis potentially closer to a

“real-world” traumatic injury and to stimulate muscle regeneration processes allowing for muscle sampling and muscle regeneration study in a more standardised way than traumatic muscle injuries (Mackey & Kjaer, 2017). To our knowledge, this is the first study that investigated the effect of CWI or HWI on muscle recovery following a muscle damage inducing myofiber necrosis.

Added in the text: “Considering the diversity of the traumatic skeletal muscle injuries, future studies should aim at examining more specifically the effect of thermal therapies in injured athletes.”

Because muscle biopsies are invasive, it is often difficult to obtain more than three or four biopsies from individuals. That said, the rationale for the timing of muscle biopsies in the present study. The authors may have missed the peak in some variables in the muscle tissue. The authors presented some histology and immunohistochemistry images. However, they did not conduct any quantitative analysis of the number/cross-sectional areas of regenerating muscle fibres. Nor did they compare the images between the treatment groups. Because muscle regeneration was the key aspect of this study, this lack of analysis is an obvious omission. It would have been useful to measure the number of leucocytes and satellite cells in the muscle biopsies as an indication of inflammation and muscle regeneration, respectively. The authors should discuss the issues as limitations of their study.

AR: Added in the limitations: “Furthermore, it should be noted that the current study lacks histochemical evaluations, such as macrophages or satellite cells quantifications in the injured muscle in this study, limiting the possible conclusions provided by this manuscript.”

Validity of the conclusions: The authors conclude overall that compared with cold water, hot water immersion may be more favourable for promoting muscle regeneration. This conclusion was based on findings that hot water immersion reduced muscle soreness and circulating muscle damage markers. Unfortunately, there is not really enough data to support the conclusion that hot water immersion reduced pro-inflammatory responses and enhanced anti-inflammatory responses. This conclusion was based only on measurements of NF-kB and IL_10, and only two biopsies at day 5 and 11.

AR: We agree with the reviewer that the supporting evidence provided to confirm HWI reduced pro-inflammatory responses and enhanced anti-inflammatory responses remain too limited to ascertain it. The abstract, discussion on NF-kB/IL-10 and the conclusion have been modified as a suggestion/hypothesis that needs future understanding.

Modified in the text: “However, further investigations are necessary to confirm the effect of HWI on the inflammatory shift.”

Conclusion: “In contrast, heat therapy could offer beneficial effects on muscle recovery and may represent a promising solution to accelerate muscle healing.”

The mention “suggesting an earlier shift from pro to anti-inflammatory phase” has been removed from the bullet points, abstract and discussion.

SPECIFIC COMMENTS:

The manuscript has quite a few grammatical and formatting errors. It would benefit from editing by a native English speaker.

AR: As requested, the manuscript has been fully reviewed by a native English speaker and corrected. Furthermore, the grammatical errors indicated by both reviewers have been corrected.

The use of different terms such as cold water immersion, COLD, cryotherapy, HWI and HOT disrupts the readability of the manuscript. It would be better to use one or two terms consistently throughout the manuscript.

AR: We thank the reviewer for his suggestion. The terms COLD, NEUTRAL and HOT, which designated the intervention groups, have been modified to cold water immersion (CWI), thermoneutral water immersion (TWI), and hot water immersion (HWI) to improve readability of the manuscript. The terms “cryotherapy” and “heat therapy” have been retained to refer to the general use of cooling and heating as injury treatments. CWI and HWI specifically refer to the intervention modalities used in this study. The figures have been modified accordingly.

In lines 157-158, the authors stated that they excluded anyone with abnormally high CK activity or myoglobin concentration. It would be useful if the authors indicated how many people were excluded on this basis.

AR: No participant reached the exclusion threshold. All participants showed normal values of CK and myoglobin for an active population before the start of the experiment. The sentence was modified to clarify the normal ranges that all participants had to ensure before the start of the experiment.

Modified in the manuscript: “Circulating Creatine-Kinase and Myoglobin levels were assessed during the week preceding the start of the experiment to ensure that all participants had values within the normal ranges (31-936U/L for CK and 0-72µg/L for myoglobin).”

As there were 34 people who completed the study, and three groups, the number of people per group should be updated (i.e., from n=12/group) throughout the manuscript.

AR: Modified in the manuscript: “i) COLD (N=12), ii) NEUTRAL (N=11), or iii) HOT (N=11)” throughout the manuscript.

Line 298: RNA extraction is mentioned, but there is no mRNA data presented in the manuscript.

AR: The mention of RNA extraction has been removed.

Figure 2 requires a scale bar on these images.

AR: The scale bars have been added to the images and the length is described in the figure 2 legend.

Line 375: This subheading doesn't see appropriate for all of the data presented in this paragraph.

AR: As recommended, the paragraph has been divided in 2 new different sections: Muscle damage and Core body and muscle temperatures.

Lines 443-444. The association between muscle soreness and VEGF expression is unclear.

AR: As suggested, the mention of VEGF has been removed from this paragraph.

Lines 462-464. This conclusion seems quite superficial.

AR: We thank the reviewer for pointing out this weakness in our manuscript.

The discussion and conclusion around pain perception and force recovery have been merged and significantly redrafted: "Although, the mechanisms promoting improvements in force recovery and pain perception using HWI remain unclear, several explanations have been proposed. As HWI increases blood flow and vasodilation (Heinonen et al., 2011; Francisco et al., 2021; Cheng et al., 2021), it has been suggested to promote healing by increasing nutrient and oxygen supplies to the injury and elicit hypoalgesia by accelerating the removal of factors sensitizing muscle nociceptors (Malanga et al., 2015; Kim et al., 2019). Indeed, mechanical hyperalgesia following an eccentric muscle damage has been partly linked to two neurotrophic factors, the nerve growth factor (NGF) and the glial cell line-derived neurotrophic factor, that may stimulate muscle nociceptors (Mizumura & Taguchi, 2016; Peake et al., 2017a). Interestingly, in rats, heat exposure using hot packs has been shown to reduce NGF muscle level and decrease physical inactivity-induced mechanical hyperalgesia (Nakagawa et al., 2018). In contrast, no significant modulation of NGF was found after a CWI following intense resistance exercise in humans (Peake et al., 2017b). Although neurotrophic factors were not measured in the current study, blood sample results indicate a potential acceleration in the removal of markers of muscle damage also possibly related to the observed hypoalgesia."

Lines 472-474. This statement is not very accurate, as there are a number of studies that have reported improvements in muscle strength following cold water immersion.

AR: This statement has been modified to be more accurate and descriptive:

"This aligns with a part of the literature showing no benefit of CWI on isometric force recovery following exercise-induced muscle damages or resistance exercise (Goodall & Howatson, 2008; Howatson et al., 2009; Crystal et al., 2013; Glasgow et al., 2014; Vieira Ramos et al., 2016; Machado et al., 2017; Sautillet et al., 2024). Interestingly, several studies reported a beneficial effect of CWI on isometric peak torque, squat jump and counter movement jump performance recovery, however, the exact mechanism remains unclear (Vaile et al., 2008; Leeder et al., 2012; Pointon et al., 2012; Roberts et al., 2015a; Chaillou et al., 2022). In contrast, chronic CWI exposure for 12 weeks following resistance training was shown to lower force and muscle mass adaptations elicited by resistance training (Roberts et al., 2015b)."

Lines 480-482. This conclusion is very vague.

AR: We thank the reviewer for pointing out this weakness in the discussion of the force results.

The discussion and conclusion around force recovery and pain have been merged and significantly redrafted and modified: "Although, the mechanisms promoting improvements in force recovery and pain perception using HWI remain unclear, several explanations have been proposed. As HWI increases blood flow and vasodilation (Heinonen et al., 2011; Francisco et al., 2021; Cheng et al., 2021), it has

been suggested to promote healing by increasing nutrient and oxygen supplies to the injury and elicit hypoalgesia by accelerating the removal of factors sensitizing muscle nociceptors (Malanga et al., 2015; Kim et al., 2019). Indeed, mechanical hyperalgesia following an eccentric muscle damage has been partly linked to two neurotrophic factors, the nerve growth factor (NGF) and the glial cell line-derived neurotrophic factor, that may stimulate muscle nociceptors (Mizumura & Taguchi, 2016; Peake et al., 2017a). Interestingly, in rats, heat exposure using hot packs has been shown to reduce NGF muscle level and decrease physical inactivity-induced mechanical hyperalgesia (Nakagawa et al., 2018). In contrast, no significant modulation of NGF was found after a CWI following intense resistance exercise in humans (Peake et al., 2017b). Although neurotrophic factors were not measured in the current study, blood sample results indicate a potential acceleration in the removal of markers of muscle damage also possibly related to the observed hypoalgesia.”

Lines 497-500. This conclusion seems quite superficial, and it doesn't attempt to tie in the data around HSPs.

AR: We thank the reviewer for pointing out this weakness in the discussion of the blood markers results. The discussion and conclusion have been redrafted to provide better understanding of our results:

“In the context of muscle injury, one of the suggested benefits of CWI is to minimize secondary cell damage (Merrick, 2002; Bleakley & Davison, 2010b). However, our results showing no significant effect of CWI on circulating markers of muscle damage (Figure 6) suggest no benefit of CWI in limiting secondary injury or improving muscle recovery from large muscle damage. Conversely, our results showed reduced levels of CK and myoglobin following HWI and suggest that HWI induced a faster removal of the markers of muscle damage transferred from the muscle to the bloodstream. Indeed, as illustrated previously in the literature, the increase in blood flow and the vasodilation elicited by the HWI is likely to have played a large role in the observed reduction in circulating markers of muscle damage (Heinonen et al., 2011; Francisco et al., 2021; Cheng et al., 2021). Moreover, HSPs have been suggested to improve cell survival and facilitate muscle recovery (Thompson et al., 2001; Paulsen et al., 2007; Fennel et al., 2022). Indeed, HSP27 has been shown to bind to cytoskeletal/myofibrillar proteins immediately following an eccentric exercise and potentially stabilize the disrupted myofibrillar structures (Paulsen et al., 2007). Moreover, in injured rats, HSP72 expression has been reported to increase following heat stress treatment (Kojima et al., 2007; Oishi et al., 2009; Shibaguchi et al., 2016) but not after local icing (20min at 0°C) (Shibaguchi et al., 2016). According to the literature, the observed HSP27 and HSP70 upregulations in HWI could have been partly responsible for the reduced CK/myoglobin levels through a reduction in the extent of muscle damage (Fennel et al., 2022). However, the reduction in CK/myoglobin levels appears earlier than the observed upregulation in HSPs. Of note, as no early muscle sample (D0 to D5) was collected, the complete kinetic of HSP remains unknown in this study”

Lines 579-584. There are some conflicting statements here that require more careful thought. How might reduced VEGF expression assist muscle regeneration?

AR: We thank the reviewer for pointing out this unclear and too speculative statement.

This statement has been modified: “Similarly to the previous human results on VEGF, no significant change in time of VEGF was observed in CWI group. In contrast, VEGF levels were significantly lower in

TWI and HWI at D11, suggesting a maintenance of pro-angiogenic factors following CWI but not following HWI.”

Lines 635-638. There are some conflicting statements here that require more careful thought.

AR: This statement has been removed from the manuscript.

Lines 641-643. This sentence is confusing.

AR: This sentence has been removed from the manuscript during the redrafting of the discussion.

The references are not formatted consistently.

AR: The references have been checked for consistency.

Because TGF-beta1 did not change, it is debatable whether it should be mentioned in the discussion.

AR: We thank the reviewer for this suggestion.

We kept a brief discussion around TGF- β 1. The absence of change remained for us a result to highlight for 2 reasons:

- Previous animal studies reported large changes in TGF- β 1 expression following various thermal interventions potentially leading to modification in extra cellular matrix regulation. However, this was not the case in our study.
- TGF- β 1 activity is influenced by many markers during the regeneration, and it appears surprising that no significant changes in time or between groups was observed, considering the different trajectories in NF- κ B and IL-10 expressions observed.

Thus, this part has been rewritten as follows:

"Despite the necessary role of TGF- β 1 during muscle regeneration, excessive elevations have been associated with fibrosis development (Delaney *et al.*, 2017). In rats, excessive collagen formation was reported following a single post-injury ice application (Takagi *et al.*, 2011; Shibaguchi *et al.*, 2016) in contrast to heat application that limited fibrosis development (Takeuchi *et al.*, 2014; Shibaguchi *et al.*, 2016). Contrary to the animal literature, we did not find any differences in TGF- β 1 expression between treatments; however, a larger variability was observed for CWI and TWI compared to HWI (**Figure 11**). Although the previous data presented for p-NF- κ B and IL-10 expressions suggest an earlier inflammatory shift in HWI only, TGF- β 1 remained weakly expressed in HWI."

The manuscript is missing some key papers that should be included for a more balanced discussion (e.g., Roberts et al. 2015 J Physiol 593: 4285-4301; Peake et al 2017 J Physiol 595: 695-711; D'Souza et al 2018 Am J Physiol 314: R824-R833; Peake et al 2020 Front Physiol 11: 737.

AR: The suggestions of the reviewer have been added to the manuscript. Furthermore, Normand-Gravier et al (2024) Eur J Appl Physiol has been added to the manuscript.

Dear Dr Racinais,

Re: JP-RP-2025-287777R1 "Muscle regeneration is improved by hot water immersion but unchanged by cold following a simulated musculoskeletal injury in humans" by Valentin Dablainville, Adele Mornas, Tom Normand-Gravier, Maha Al-Mulla, Emmanouil Papakostas, Bruno Olory, Theodorakys Marín Fermín, Frantzeska Zampeli, Nelda Nader, Marine Alhammoud, Freya Bayne, Anthony Sanchez, Marco Cardinale, Robin Candau, Henri Bernardi, and Sebastien Racinais

Thank you for submitting your manuscript to The Journal of Physiology. It has been assessed by a Reviewing Editor and by 2 expert referees and we are pleased to tell you that it is acceptable for publication following satisfactory revision.

REVISION CHECKLIST:

We look forward to receiving your revised submission.

Yours sincerely,

Paul Greenhaff
Senior Editor
The Journal of Physiology

EDITOR COMMENTS

Reviewing Editor:

Ethics Concerns:

The study could not recruit enough females and was performed only on males.

Comments to the Author:

The authors are encouraged to mitigate the ethical concern related to the exclusion of female participants by acknowledging this limitation and discussing any biological or clinical relevance of potential sex differences. The authors should also propose strategies for future studies that promote more inclusive participation, such as multi site collaboration, targeted recruitment, etc. The authors should continue to revise the manuscript based on the suggestions of reviewers 1 and 2.

Senior Editor:

Comments to the Author:

This revised manuscript has been considered by the same reviewing editor and reviewers that considered the original submission. All are of the opinion that the manuscript has been improved. However, both reviewers have raised a number of specific points that require further consideration by the authors. The Reviewing Editor has also requested in their feedback that the authors mitigate the ethical concern related to the exclusion of female participants and helpfully has suggested a number of ways to achieve this. We look forward to receiving the further revised manuscript.

REFEREE COMMENTS

Referee #1:

Thanks for the opportunity to review this subsequent version of the manuscript. The gracious acceptance of the previous feedback is appreciated.

Specific comments

Line 158-159: This sentence might be clearer if reworded to 'The sample size was increased to 12/group to compensate for potential participant drop-out' or similar.

Line 161: By "recreative", do you mean 'recreational'?

Lines 163-168: The described exclusion of the female participant raises concerns. Historically, women have been underrepresented in research, leading to gaps in knowledge about female health and responses to treatments. Excluding the female participant perpetuates this issue and undermines efforts to increase inclusivity in research. The evidence referenced relating to sex differences and muscle regeneration is acknowledged. However, various statistical methods exist to accommodate for sex differences in analyses. Adopting such an approach would have increased the generalizability of the findings. Excluding participants based on sex, especially when sex is known to affect the outcomes, can be seen as unethical and discriminatory. Ensuring that the study design is inclusive and that the results apply to both sexes is important.

It is understood that considerable challenges exist if this situation were to be overcome. Accordingly, this view will be left to the editor to judge.

Line 181-184: Ensuring participant compliance with instructions is difficult when not directly under supervision. However, adherence to dietary instructions can be examined quickly and cheaply using compliance questionnaires, self-report diaries, photographic records, or food-tracking apps. Diet can considerably affect muscle recovery; thus, the absence of this check is a limitation that should be acknowledged.

Discussion: Although lengthy, the edits made to the Discussion have improved it greatly.

Referee #2:

Thank you for attending to my comments on the previous version of this manuscript. The revisions you've made have improved the quality of the manuscript. Some further minor corrections are required, as outlined below:

L74. Change 'condition' to 'conditions'.

L158. Change to 'compensate for possible withdrawal of participants'.

L161. Change to 'recreational'.

L166. 'Repartition' is not an English word. I think you mean 'representation'.

L168. Change to 'creatine' and 'myoglobin' (with lower case letters).

L170. Add a space between the numbers and units.

L173. Change 'condition' to 'conditions'.

L183. Change to 'supplements'.

L282. Delete 'a' before 'Biopac'.

L313. Change 'experiment' to 'study period'.

L315-317. Delete 'unlike a recent study' and the reference at the end of the sentence. It's not necessary to refer to this other research.

L340. Change to 'before being analysed'.

L423-424. A brief description of notable features in Figure 2 is needed here.

L464. This subheading seems a bit odd for the results. Perhaps Protein expression of heat shock proteins etc etc. would be more appropriate for the results.

L500. '...a large muscle damage inducing myofiber necrosis' is poorly worded and difficult to understand. Please revise, and change to UK spelling for 'myofibre'.

L510. Change to 'damage'.

L515. Change to 'mechanisms remain'.

L521. It's unclear what you mean by 'muscle mass recovery'.

L527. Change 'have' to 'has'. Also, what does the word 'some' refer to here? Studies?

L529. What does the word 'others' refer to here? Studies?

L530. Change 'it' to 'CWI'.

L532. Change to 'damage'.

L534-535. '...multiple 20 to 30 minutes' is poorly worded. Please revise.

L550. Change to 'supply'.

L551. Change to 'sensitising'.

L556. Change to 'expression levels'.

L572. Change to 'minimise'.

L577-578. The wording 'faster...bloodstream' does not really make sense. Please revise.

L581-592. I'm not sure this discussion about HSPs really fits in this particular paragraph. It probably fits better in the

paragraph below about muscle regeneration.

L620. Change to 'a water perfused suit'.

L626-627. The wording 'only...HWI' does not really make sense. Please revise.

L659. Change to 'terms'.

L678. Change to 'did not allow us to detect'.

L701. Change 'that' to 'which'.

L721. Change to 'did not allow us to determine'.

L723-724. Change to 'Although very unlikely, it is not possible to determine whether hydrostatic pressure influenced muscle recovery...!'.

L725-726. The wording 'Moreover, the effect of water temperature being larger on peripheral muscle tissue than on the deeper level...' does not make sense. Please revise.

L750-751. One sentence does not constitute a paragraph. Please add another 1-2 sentences to expand on this point.

The titles of some of the references require formatting so that they are presented in 'sentence case', with only the first word of the title starting with an upper case letter.

END OF COMMENTS

Reviewing Editor:

Ethics Concerns:

The study could not recruit enough females and was performed only on males.

Comments to the Author:

The authors are encouraged to mitigate the ethical concern related to the exclusion of female participants by acknowledging this limitation and discussing any biological or clinical relevance of potential sex differences. The authors should also propose strategies for future studies that promote more inclusive participation, such as multi site collaboration, targeted recruitment, etc. The authors should continue to revise the manuscript based on the suggestions of reviewers 1 and 2.

AR: The authors thank the reviewing editor for the positive feedback and useful suggestions. We have integrated all reviewer suggestions and hope that the manuscript is now clearer. The modifications are indicated in the revised text in red font and our replies are included bellow.

Regarding the recruitment of female participants, it has been clarified in the text that: "...this research project was open to male and female participants. However, only one female volunteered to participate and could not be included as she was not matching the criteria for group counterbalancing. This is an unfortunate limitation as there are large differences in muscle regeneration recently reported between sexes (Luk et al., 2021; You et al., 2023). While this limitation could be partly related to the local culture where the study was conducted, it is consistently more difficult to include women when the study may leave a scar due to muscle biopsies. Future studies are encouraged to include more women to participate by promoting more inclusive protocol or encouraging multi-site collaboration."

Senior Editor:

Comments to the Author:

This revised manuscript has been considered by the same reviewing editor and reviewers that considered the original submission. All are of the opinion that the manuscript has been improved. However, both reviewers have raised a number of specific points that require further consideration by the authors. The Reviewing Editor has also requested in their feedback that the authors mitigate the ethical concern related to the exclusion of female participants and helpfully has suggested a number of ways to achieve this. We look forward to receiving the further revised manuscript.

AR: The authors thank the senior editor for the opportunity to amend this manuscript. We have integrated all reviewer suggestions and hope that the manuscript is now clearer.

Referee #1:

Thanks for the opportunity to review this subsequent version of the manuscript. The gracious acceptance of the previous feedback is appreciated.

AR: The authors thank the reviewer for the useful suggestions. The modifications are indicated in red font and our replies are included below.

Specific comments

Line 158-159: This sentence might be clearer if reworded to 'The sample size was increased to 12/group to compensate for potential participant drop-out' or similar.

AR: The sentence has been reworded as suggested.

Line 161: By "recreative", do you mean 'recreational'?

AR: Corrected in the text.

Lines 163-168: The described exclusion of the female participant raises concerns. Historically, women have been underrepresented in research, leading to gaps in knowledge about female health and responses to treatments. Excluding the female participant perpetuates this issue and undermines efforts to increase inclusivity in research. The evidence referenced relating to sex differences and muscle regeneration is acknowledged. However, various statistical methods exist to accommodate for sex differences in analyses. Adopting such an approach would have increased the generalizability of the findings. Excluding participants based on sex, especially when sex is known to affect the outcomes, can be seen as unethical and discriminatory. Ensuring that the study design is inclusive and that the results apply to both sexes is important. It is understood that considerable challenges exist if this situation were to be overcome. Accordingly, this view will be left to the editor to judge.

AR: We thank the reviewer for his comment. We understand and agree with his point. However, only one female potential participant showed an interest to participate during the 6 months of recruitment for this research project. This participant was not excluded but the criteria to equally counterbalance the groups were not met by including only one female participant.

This section was moved as a limitation of this study.

In the text: "Of note, this research project was open to male and female participants. However, only one female volunteered to participate and could not be included as she was not matching the criteria for group counterbalancing. This is an unfortunate limitation as there are large differences in muscle regeneration recently reported between sexes (Luk et al., 2021; You et al., 2023). While this limitation could be partly related to the local culture where the study was conducted, it is consistently more difficult to include women when the study may leave a scar due to muscle biopsies. Future studies are encouraged to include more women to participate by promoting more inclusive protocol or encouraging multi-site collaboration."

Line 181-184: Ensuring participant compliance with instructions is difficult when not directly under supervision. However, adherence to dietary instructions can be examined quickly and cheaply using compliance questionnaires, self-report diaries, photographic records, or food-tracking apps. Diet can considerably affect muscle recovery; thus, the absence of this check is a limitation that should be acknowledged.

AR: This point has been added as a limitation of this study.

In the text: "Although participants received nutritional interventions from a sport dietician and were reminded several times by the research team, their compliance with the nutritional instructions was not monitored."

Discussion: Although lengthy, the edits made to the Discussion have improved it greatly.

AR: Thank you.

Referee #2:

Thank you for attending to my comments on the previous version of this manuscript. The revisions you've made have improved the quality of the manuscript. Some further minor corrections are required, as outlined below:

AR: The authors thank the reviewer for its useful suggestions. The references have been reformatted. The modifications are indicated in red font and our replies are included below.

L74. Change 'condition' to 'conditions'.

AR: Modified by “conditions” in the text.

L158. Change to 'compensate for possible withdrawal of participants'.

AR: Modified in the text following reviewer 1 suggestion.

In the text: “The sample size was increased to 12/group to compensate for potential participant drop-out.”

L161. Change to 'recreational'.

AR: Modified by “recreational” in the text.

L166. 'Repartition' is not an English word. I think you mean 'representation'.

AR: Modified by “representation” in the text.

L168. Change to 'creatine' and 'myoglobin' (with lower case letters).

AR: Creatine and myoglobin have been modified with lower case letters in the text.

L170. Add a space between the numbers and units.

AR: spaces have been added.

L173. Change 'condition' to 'conditions'.

AR: Modified by “conditions” in the text.

L183. Change to 'supplements'.

AR: Modified by “supplements” in the text.

L282. Delete 'a' before 'Biopac'.

AR: “a” has been removed.

L313. Change 'experiment' to 'study period'.

AR: modified by “study period” in the text.

L315-317. Delete 'unlike a recent study' and the reference at the end of the sentence. It's not necessary to refer to this other research.

AR: Adding this reference was suggested by reviewer 1 during the 1st round of review. This reference has been moved and added to the limitation section of the manuscript.

In the text: “Future studies should include a measurement of subcutaneous fat tissue and muscle thickness to improve the precision of the thermal intervention (Rodrigues *et al.*, 2024).”

L340. Change to 'before being analysed'.

AR: Modified in the text.

L423-424. A brief description of notable features in Figure 2 is needed here.

AR: A brief description has been added in the results. The figure is also described in the figure caption.

In the results section: “As shown in Figure 2, myofibres infiltrated by nuclei and negative to dystrophin staining became visible following the simulated injury thus confirming myofibres undergoing cells necrosis.”

In the figure caption: “Fig. 2: Microscope images of histologic staining illustrating the effect of the muscle damage protocol at D5 and D11 in comparison to PRE. H&E, laminin (green), nuclei (blue), and dystrophin (orange) staining are displayed. The symbol (*) identifies the same fibre across the different stains for each sampling time. Examples of damaged fibres are identified by arrows at D5 and D11, these fibres present a swollen aspect with a loss of normal polygonal outline, mononuclear cells infiltration, internal nuclei, or a loss of dystrophin immunoreactivity of their membrane. A large number of similar fibres presenting necrosis are visible in the D11 illustration. Scale bars, 50 µm.”

L464. This subheading seems a bit odd for the results. Perhaps Protein expression of heat shock proteins etc etc. would be more appropriate for the results.

AR: Subheading has been modified in the text as suggested.

L500. '...a large muscle damage inducing myofiber necrosis' is poorly worded and difficult to understand. Please revise, and change to UK spelling for 'myofibre'.

AR: Modified in the text by: “This is the first study to examine the effects of CWI and HWI on muscle regeneration following an electrically stimulated eccentric exercise.”

L510. Change to 'damage'.

AR: modified in the text.

L515. Change to 'mechanisms remain'.

AR: modified in the text.

L521. It's unclear what you mean by 'muscle mass recovery'.

AR: For more accuracy the terminology “Muscle mass recovery” has been modified in the text by “attenuating hypertrophy signalling”.

In the text: “If our results reported neither benefit nor detrimental effect of a relatively short-term CWI on force recovery, based on the existing literature, it could be hypothesised that chronic CWI used post-injury might interfere with rehabilitation processes by reducing force recovery and attenuating hypertrophy signalling.”

L527. Change 'have' to 'has'. Also, what does the word 'some' refer to here? Studies?

AR: Modified to “has” in the text.

Some was referring to “studies”. It has been modified for more clarity.

In the text: “However, equivocal evidence regarding the effect of CWI on muscle pain and soreness following exercise-induced muscle damage has been reported. A part of the literature reports reductions (Leeder et al., 2012; Machado et al., 2016; Siqueira et al., 2018; Hohenauer et al., 2020; Moore et al., 2022; Huang et al., 2024), while in contrary, several studies report no significant benefit or detrimental effects of CWI (Sellwood et al., 2007; Goodall & Howatson, 2008; Vaile et al., 2008; Howatson et al., 2009; Glasgow et al., 2014; Sautillet et al., 2024).”

L529. What does the word 'others' refer to here? Studies?

AR: modified in the text for more clarity.

In the text: “A part of the literature reports reductions in muscle pain and soreness (Leeder et al., 2012; Machado et al., 2016; Siqueira et al., 2018; Hohenauer et al., 2020; Moore et al., 2022; Huang et al., 2024), while in contrary, several studies report no significant benefit or detrimental effects of CWI (Sellwood et al., 2007; Goodall & Howatson, 2008; Vaile et al., 2008; Howatson et al., 2009; Glasgow et al., 2014; Sautillet et al., 2024).”

L530. Change 'it' to 'CWI'.

AR: Modified in the text.

L532. Change to 'damage'.

AR: Modified in the text.

L534-535. '...multiple 20 to 30 minutes' is poorly worded. Please revise.

AR: It has been reworded.

In the text: “To our knowledge, the only study that examined the effect of cryotherapy on muscle tear injury, showed no benefit of chronic icing (20 to 30 minutes applications) during the first 36 hours post-injury on muscle pain perception at rest or during physical activity (Prins et al., 2011).”

L550. Change to 'supply'.

AR: Modified in the text.

L551. Change to 'sensitising'.

AR: Modified in the text.

L556. Change to 'expression levels'.

AR: Modified in the text.

L572. Change to 'minimise'.

AR: Modified in the text.

L577-578. The wording 'faster...bloodstream' does not really make sense. Please revise.

AR: This sentence has been reworded.

In the text: "Conversely, our results showed reduced levels of CK and myoglobin following HWI, suggesting that HWI potentially increased the rate at which CK/myoglobin were removed from the muscle and entered the bloodstream."

L581-592. I'm not sure this discussion about HSPs really fits in this particular paragraph. It probably fits better in the paragraph below about muscle regeneration.

AR: We thank the reviewer for this suggestion. The paragraph on HSPs impact on CK/myoglobin was moved with the rest of the discussion about HSPs in the muscle regeneration section.

In the text: "While the precise role of HSPs in post-injury recovery remains unclear, the literature indicates that HSPs may have played a key role in protecting and facilitating skeletal muscle regeneration. HSPs have been suggested to improve cell survival and facilitate muscle recovery (Thompson et al., 2001; Paulsen et al., 2007; Fennel et al., 2022). Indeed, HSP27 has been shown to bind to cytoskeletal/myofibrillar proteins immediately following an eccentric exercise and potentially stabilise the disrupted myofibrillar structures (Paulsen et al., 2007). In injured rats, HSP72 expression has been reported to increase following heat stress treatment (Kojima et al., 2007; Oishi et al., 2009; Shibaguchi et al., 2016) but not after local icing (20min at 0°C) (Shibaguchi et al., 2016). According to the literature, the observed HSP27 and HSP70 upregulations in HWI could have been partly responsible for the reduced CK/myoglobin levels through a reduction in the extent of muscle damage (Fennel et al., 2022). However, the reduction in CK/myoglobin levels appears earlier than the observed upregulation in HSPs."

L620. Change to 'a water perfused suit'.

AR: Modified in the text.

L626-627. The wording 'only...HWI' does not really make sense. Please revise.

AR: The sentence has been reworded.

In the text: "If high core temperatures might elicit additional benefits, in our study, the large use of cooling solutions on the upper body limited core temperature to 38°C at the end of HWI."

L659. Change to 'terms'.

AR: Modified in the text.

L678. Change to 'did not allow us to detect'.

AR: Modified in the text.

L701. Change 'that' to 'which'.

AR: Modified in the text.

L721. Change to 'did not allow us to determine'.

AR: Modified in the text.

L723-724. Change to 'Although very unlikely, it is not possible to determine whether hydrostatic pressure influenced muscle recovery...!.

AR: Modified in the text.

L725-726. The wording 'Moreover, the effect of water temperature being larger on peripheral muscle tissue than on the deeper level...' does not make sense. Please revise.

AR: this sentence has been reworded.

In the text: "Water immersions induced larger modifications in temperature in peripheral tissue (1cm depth) than in deep tissue (3cm depth) (Figure 3). However, all tissue depths were impacted by the thermal treatments, as well as central body temperature."

L750-751. One sentence does not constitute a paragraph. Please add another 1-2 sentences to expand on this point.

AR: This point has been moved and expanded in the text.

In the text: "The injury model used in this study induced a large myofibre necrosis, which allowed for muscle sampling and the investigation of muscle regeneration processes in a more standardised way compared to other traumatic skeletal muscle injury models (Mackey & Kjaer, 2017). Considering the differences between traumatic skeletal muscle injuries, it remains unclear how well our findings on thermal therapies apply to other types of muscle injuries (Edouard et al., 2023). Future studies should aim at examining more specifically the effect of thermal therapies in injured athletes."

The titles of some of the references require formatting so that they are presented in 'sentence case', with only the first word of the title starting with an upper case letter.

AR: Modified in the references.

Dear Dr Racinais,

Re: JP-RP-2025-287777R2 "Muscle regeneration is improved by hot water immersion but unchanged by cold following a simulated musculoskeletal injury in humans" by Valentin Dablainville, Adele Mornas, Tom Normand-Gravier, Maha Al-Mulla, Emmanouil Papakostas, Bruno Olory, Theodorakys Marín Fermín, Frantzeska Zampeli, Nelda Nader, Marine Alhammoud, Freya Bayne, Anthony Sanchez, Marco Cardinale, Robin Candau, Henri Bernardi, and Sebastien Racinais

Thank you for submitting your manuscript to The Journal of Physiology. It has been assessed by a Reviewing Editor and by 2 expert referees and we are pleased to tell you that it is acceptable for publication following satisfactory revision.

REVISION CHECKLIST:

We look forward to receiving your revised submission.

Yours sincerely,

Paul Greenhaff
Senior Editor
The Journal of Physiology

EDITOR COMMENTS

Reviewing Editor:

Comments to the Author:

The authors have adequately addressed all comments. Both reviewers are satisfied with the revised manuscript and recommend it for publication. However, reviewer 2 has suggested a few minor corrections that should be addressed prior to final acceptance.

Senior Editor:

Comments to the Author:

Thank you for the revised manuscript that was considered by the same reviewing editor and reviewers that considered the original submission. All are in agreement that the manuscript has been improved and is acceptable for publication. Reviewer 2 however has a list of minor edits that they believe need to be addressed by the authors. The manuscript can be accepted with the proviso that the authors make the requested minor changes. Thank you for considering The Journal of Physiology to publish your research.

REFEREE COMMENTS

Referee #1:

Thanks for your considered response to my feedback. I have no further suggestions. Congratulations on your paper.

Referee #2:

Thank you for addressing my previous comments. A few more minor corrections are required, as outlined below:

L418. Change to 'cellular necrosis'.

L529. Change 'contrary' to 'contrast'.

L534. Change to '20- to 30-minute applications'.

L754. Change 'was not matching' to 'did not match'.

L1199, 1206, 1210, 1215, 1220 and 1225. Please change to 'along with individual data'.

END OF COMMENTS

Reviewing Editor:

Comments to the Author:

The authors have adequately addressed all comments. Both reviewers are satisfied with the revised manuscript and recommend it for publication. However, reviewer 2 has suggested a few minor corrections that should be addressed prior to final acceptance.

AR: The authors thank the reviewing editor for the positive feedback and useful suggestions. We have integrated all reviewer suggestions and hope that the manuscript is now clearer. The modifications are indicated in the revised text in red font and our replies are included below.

Senior Editor:

Comments to the Author:

Thank you for the revised manuscript that was considered by the same reviewing editor and reviewers that considered the original submission. All are in agreement that the manuscript has been improved and is acceptable for publication. Reviewer 2 however has a list of minor edits that they believe need to be addressed by the authors. The manuscript can be accepted with the proviso that the authors make the requested minor changes. Thank you for considering The Journal of Physiology to publish your research.

AR: The authors thank the senior editor for the positive feedback and the opportunity to amend this manuscript. We have integrated all reviewer suggestions and hope that the manuscript is now clearer.

Referee #2:

Thank you for addressing my previous comments. A few more minor corrections are required, as outlined below:

AR: The authors thank the reviewer for its positive feedback and suggestions during the reviewing process that improved the manuscript. The modifications are indicated in red font and our replies are included below.

L418. Change to 'cellular necrosis'.

AR: Modified to "cellular necrosis"

L529. Change 'contrary' to 'contrast'.

AR: Modified to "in contrast"

L534. Change to '20- to 30-minute applications'.

AR: Modified to "20- to 30-minute applications"

L754. Change 'was not matching' to 'did not match'.

AR: Modified to "did not match"

L1199, 1206, 1210, 1215, 1220 and 1225. Please change to 'along with individual data'.

AR: Modified to "along with individual data"

Dear Dr Racinais,

Re: JP-RP-2025-287777R3 "Muscle regeneration is improved by hot water immersion but unchanged by cold following a simulated musculoskeletal injury in humans" by Valentin Dablainville, Adele Mornas, Tom Normand-Gravier, Maha Al-Mulla, Emmanouil Papakostas, Bruno Olory, Theodorakys Marín Fermín, Frantzeska Zampeli, Nelda Nader, Marine Alhammoud, Freya Bayne, Anthony Sanchez, Marco Cardinale, Robin Candau, Henri Bernardi, and Sebastien Racinais

We are pleased to tell you that your paper has been accepted for publication in The Journal of Physiology.

Yours sincerely,

Paul Greenhaff
Senior Editor
The Journal of Physiology

If you would like to receive our 'Research Roundup', a monthly newsletter highlighting the cutting-edge research published in The Physiological Society's family of journals (The Journal of Physiology, Experimental Physiology, Physiological Reports, The Journal of Nutritional Physiology and The Journal of Precision Medicine: Health and Disease), please click this link, fill in your name and email address and select 'Research Roundup':
<https://www.physoc.org/journals-and-media/membernews>

- You can help your research get the attention it deserves! Check out Wiley's free Promotion Guide for best-practice recommendations for promoting your work at: www.wileyauthors.com/eeo/guide. You can learn more about Wiley Editing Services which offers professional video, design, and writing services to create shareable video abstracts, infographics, conference posters, lay summaries, and research news stories for your research at: www.wileyauthors.com/eeo/promotion.

EDITOR COMMENTS

Senior Editor:

Comments to the Author:

Thank you for making the requested minor edits. The manuscript is now acceptable for publication. Congratulations and thank you for considering the Journal of Physiology to publish your work.